# Lenacapavir-induced capsid damage uncovers HIV-1 genomes emanating from nuclear speckles

Thorsten G Müller [1,6 ✉], Severina Klaus[1], Vojtech Zila [1,7], Bojana Lucic[2,3], Carlotta Penzo[2], Svenja L Nopper[1], Gonen Golani [4,5,8], Maria Anders-Össwein[1], Vera Sonntag-Buck[1], Anke-Mareil Heuser[1], Ulrich S Schwarz [4,5], Vibor Laketa[1,3], Marina Lusic [2,3], Barbara Müller [1] & Hans-Georg Kräusslich [1,3 ✉]

## Abstract

**Following cell entry, HIV-1 capsids enter the nucleus by passage through nuclear pores and reach nuclear speckles with subsequent uncoating of the reverse-transcribed genome and its integration into speckle-associated chromatin domains. Here, we characterized the ultrastructure of HIV-1 subviral complexes in nuclei of primary monocyte-derived macrophages and cell lines using live-cell imaging, super-resolution microscopy, and correlative light and electron tomography in the absence and presence of capsid-targeting inhibitors Lenacapavir and PF74. Capsid-like structures containing viral DNA, as well as broken capsids, clustered in nuclear speckles and were displaced from speckles by drug treatment. This was accompanied by alteration of the nuclear capsid structure, with electron-dense protrusions emanating from the narrow end of capsid cones and exposure of integration-competent genomic HIV-1 DNA. Our data indicate that synthesis of genomic dsDNA can be completed inside the closed HIV-1 capsid, and speckle-associated factors could regulate genome uncoating. This may ensure that genome uncoating occurs at optimal sites for integration into transcriptionally active chromatin. The results also shed further light on the mechanism of action of Lenacapavir.**

**Keywords** HIV, Capsid; Lenacapavir; Nuclear Speckles; Reverse Transcription; Uncoating
**Subject Categories** Chromatin, Transcription & Genomics; Microbiology, Virology & Host Pathogen Interaction; Organelles

## Introduction

Human immunodeficiency virus type 1 (HIV-1) replicates in CD4[+] T cells and non-dividing macrophages of infected individuals.

Within the last three decades, pharmacologically inhibiting viral enzymes has transformed a deadly infection into a life-long chronic condition, yet several challenges, including resistant mutations, drug adherence, and availability issues, have prevented global elimination. The recent discovery, that the viral structural capsid protein (CA) shell plays a central role for the intertwined processes essential for replication (reviewed in (Müller et al, 2022; Jang and Engelman, 2023; Morling et al, 2025)) accelerated tackling some of these challenges with a novel class of drugs (Blair et al, 2010; Bester et al, 2020; Link et al, 2020) called capsid inhibitors. One of these capsid inhibitors termed Lenacapavir (LEN, GS-6207, or brand name "Sunlenca") (Bester et al, 2020; Link et al, 2020) was recently approved by the European Medicines Agency and U.S. Food and Drug Administration for use in patients with multi-drug resistant HIV infections (Paik, 2022), and shows unprecedented efficacy in pre-exposure prophylaxis (Bekker et al, 2024), culminating in the selection of LEN as the "2024 Breakthrough of the Year" by Science magazine (Cohen, 2024).

The conical HIV-1 capsid is composed of a lattice comprising ~250 hexamers and 12 pentamers of CA, with its unique structure playing a key role in many of its functions (Sundquist and Kräusslich, 2012; Freed, 2015). Upon arrival in the host cell by fusion of the viral membrane with the plasma membrane, the capsid shell facilitates transport towards the nucleus and shields the viral genome from the innate immune system. Simultaneously, the capsid acts as a reaction container for reverse transcription, converting the viral single-stranded RNA genome into double-stranded DNA (dsDNA). This process starts following entry of the capsid into the cytoplasm (Hu and Hughes, 2012), but is only completed inside the nucleus (Dharan et al, 2020; Selyutina et al, 2020; Burdick et al, 2020; Francis et al, 2020; Rensen et al, 2021; Müller et al, 2021). The cytoplasmic capsid encasing the active replication complex enters the nucleus, where the viral dsDNA genome is released from the capsid in a process called uncoating (Dharan et al, 2020; Burdick et al, 2020; Selyutina et al, 2020; Müller et al, 2021, 2022; Zila et al, 2021; Gifford and Melikyan, 2024; Kreysing et al, 2025). The capsid integrity appears

[1]Department of Infectious Diseases, Virology, Heidelberg University, Heidelberg 69120, Germany. [2]Department of Infectious Diseases, Integrative Virology, Heidelberg University, Heidelberg 69120, Germany. [3]German Center for Infection Research, Partner Site Heidelberg, Heidelberg 69120, Germany. [4]Institute for Theoretical Physics, Heidelberg University, Heidelberg 69120, Germany. [5]BioQuant, Heidelberg University, Heidelberg 69120, Germany. [6]Present address: Max Planck Institute of Immunobiology and Epigenetics, Freiburg im Breisgau 79108, Germany. [7]Present address: Department of Capsid Development and AAV Biology, AskBio GmbH, Heidelberg 69123, Germany. [8]Present address: Department of Physics, University of Haifa, Haifa 3498838, Israel. ✉E-mail: muellert@ie-freiburg.mpg.de; Hans-Georg.Kraeusslich@med.uni-heidelberg.de

to stay mostly intact until shortly before genome integration (Li et al, 2021), which preferentially occurs into nuclear speckle-associated chromatin domains (Sowd et al, 2016; Achuthan et al, 2018; Francis et al, 2020), to ultimately drive expression of the viral proteins and ensure long-term persistence.

Nuclear entry requires transport of the viral genome and associated proteins through nuclear pore complexes (NPC). Recent evidence has shown that seemingly intact capsids enter the nucleus through nuclear pores in different cell types (Zila et al, 2021; Müller et al, 2021; Schifferdecker et al, 2022), including primary human monocyte-derived macrophages (MDM) (Kreysing et al, 2025). This requires interactions between a hydrophobic binding cleft within the capsid lattice and FG-nucleoporins (Nups) of the NPC (Fu et al, 2024; Xue et al, 2023; Dickson et al, 2024). Being the largest nuclear import cargo known to date, capsid nuclear entry represents a sterical challenge, which can induce damage to the NPC (Kreysing et al, 2025; Hou et al, 2025), with the elastic properties of its lattice structure proposed to contribute to maintaining capsid integrity (Deshpande et al, 2024). At the nucleoplasmic side of the NPC, the host factor cleavage and polyadenylation specificity factor subunit 6 (CPSF6) binds to the same binding cleft in the capsid lattice as FG-Nups and mediates capsid release from the NPC (Bejarano et al, 2019; Achuthan et al, 2018; Chin et al, 2015; Zila et al, 2019). Subsequently, capsids migrate to nuclear speckles (Francis et al, 2020; Li et al, 2020; Selyutina et al, 2020; Rohlfes et al, 2025). Clustering of incoming viral structures has been observed inside the nucleus of tissue-culture-adapted cell lines (Rensen et al, 2021; Schifferdecker et al, 2022; Müller et al, 2021).

The molecular mechanisms driving uncoating and destabilization of the capsid shell are still not completely understood. A recent report suggested that completion of reverse transcription resulting in production of long viral cDNA induces uncoating (Burdick et al, 2024). The additional presence in the capsid and the relatively rigid nature of dsDNA (Garcia et al, 2007)—in addition to the more flexible RNA (Chen et al, 2012)—may exert a disruptive mechanical outward force on the capsid. Accordingly, the genome length of HIV-1-based vectors correlated with transduction efficiency, indicating a critical minimal genome length of ~6 kb for efficient uncoating (Burdick et al, 2024). This model gained support from theoretical analyses (Rouzina and Bruinsma, 2014) and in vitro studies, in which isolated HIV-1 capsids were analyzed by atomic force microscopy (Rankovic et al, 2017, 2018, 2021; Xu et al, 2020) or cryo-EM (Christensen et al, 2020) after undergoing endogenous reverse transcription in vitro. While these results clearly point to an important role of genome length for uncoating, it is currently not clear whether completion of reverse transcription of the HIV-1 genome is sufficient or whether additional uncoating factors are required.

LEN (Bester et al, 2020; Link et al, 2020) as well as the previously described compound PF74 (Blair et al, 2010) inhibit viral infection in the early phase with multiple modes of action. These compounds bind to the same hydrophobic cleft between adjacent capsid hexamers as FG-Nups and CPSF6 (Bhattacharya et al, 2014; Bester et al, 2020). Compound binding stabilizes the hexameric lattice, but is incompatible with CA pentamers (Huang et al, 2025; Bhattacharya et al, 2014; Márquez et al, 2018; Faysal et al, 2024). This is consistent with experimental findings that PF74 or LEN addition to capsids carrying an internal fluid phase marker led to capsid

breakage and release of the marker in vitro, while most of the hexameric capsid lattice remained intact (Márquez et al, 2018; Faysal et al, 2024; Li et al, 2025). Drug binding also competitively inhibits capsid interaction with FG-repeats of Nups and CPSF6 (Bhattacharya et al, 2014; Bester et al, 2020), and therefore blocks nuclear entry and subsequent nucleoplasmic trafficking to nuclear speckles. These effects occur at much lower concentrations of LEN or PF74 than lattice disruption. Furthermore, PF74 has been shown to displace subviral HIV-1 complexes from nuclear speckles (Francis et al, 2020) and to reveal CA epitopes for immunostaining (Müller et al, 2021), both features dependent on CPSF6 coating of the capsid.

While much is known about drug binding and its effect on capsid structure and nuclear entry, the effects on subviral HIV-1 complexes inside the nucleus and on the ultrastructure of the nuclear capsid are not well understood. Here, we analyzed HIV-1 capsids and their retention within nuclear speckles in the absence or presence of LEN/PF74 in MDM and cell lines using correlative light and electron microscopy (CLEM), electron tomography (ET), and (super-resolution) fluorescence microscopy in combination with live-cell detection of viral DNA. For visualization of reverse-transcribed viral genomes, we employed either labeling of newly synthesized DNA by the nucleoside analog 5-ethynyl-2′-deoxyuridine (EdU) (Peng et al, 2015) or specific detection of HIV-1 dsDNA by the ANCHOR system (Saad et al, 2014). The latter approach is based on the prokaryotic chromosomal partitioning system ParB-*parS* and requires almost complete reverse transcription of the HIV-1 genome as well as accessibility of the viral dsDNA for detection (Müller et al, 2021).

We observed rapid exposure of previously hidden pre-synthesized nuclear HIV-1 genomic dsDNA upon LEN treatment of infected cells, which coincided with ultrastructural alterations, including bifurcated protrusions at the narrow end of capsid cones displaced from nuclear speckles. Our observations indicate that synthesis of integration-competent, functionally complete viral dsDNA can occur within the CPSF6-coated nuclear HIV-1 capsid or a capsid-like structure without immediate capsid breakage, thereby suggesting that additional factors, besides dsDNA synthesis, may be needed to trigger uncoating.

## Results

### HIV-1 capsid structures cluster in nuclear speckles of primary monocyte-derived macrophages

We first aimed to characterize the fate of HIV-1 capsids beyond the nuclear pore in terminally differentiated primary human MDM. We have previously observed clustering of HIV-1 capsids in the nuclei of HeLa-derived cell lines (Müller et al, 2021; Schifferdecker et al, 2022), and in the SupT1 T-cell line (Zila et al, 2021; Schifferdecker et al, 2022). In MDM, HIV-1 clusters of cDNA and fluorescently labeled IN largely colocalizing with nuclear speckles have been reported by others (Rensen et al, 2021; Francis et al, 2020). In order to determine whether these genome clusters also correspond to accumulations of capsids or capsid-like structures in primary MDM, we employed super-resolution fluorescence microscopy and CLEM to characterize the ultrastructure of subviral complexes associated with nuclear speckles.

Analyses were performed using an HIV-1 NL4-3 derivative carrying a deletion in the regulatory protein Tat and point mutations in the integrase (IN) active site (NNHIV; (Zila et al, 2021; Müller et al, 2021)). This variant is not replication competent, but undergoes all post-entry events before the integration stage. For detection of subviral HIV-1 complexes, we employed a fluorescent Vpr-IN fusion protein provided in trans during particle production (Albanese et al, 2008), resulting in incorporation of labeled IN into the capsid. In previous studies, labeled IN was found to remain associated with capsid-like structures in the cytosol and nucleus in various cell types (Zila et al, 2021; Müller et al, 2021), and stayed associated with structures resembling capsid remnants upon separation from the viral cDNA (Müller et al, 2021). Reverse transcribing HIV-1 replication complexes in infected cells were detected by incorporation of the nucleoside analog EdU into the nascent viral DNA, followed by click-labeling. Although EdU incorporation is non-specific, this strategy labels almost exclusively viral replication complexes inside the nuclei of terminally differentiated cells lacking cellular DNA synthesis (Peng et al, 2015; Bejarano et al, 2019; Stultz et al, 2017; Francis et al, 2020; Rensen et al, 2021; Müller et al, 2021). Nuclear speckles were identified by immunostaining with the antibody SC35, which recognizes the major speckle protein SRRM2 (Ilik et al, 2020).

MDM infected with the labeled virus were stained, and nuclear subviral complexes were imaged at 72 h post infection (p.i.) by Airyscan confocal microscopy. This analysis revealed a clear colocalization of IN.eGFP and EdU punctae within SRRM2-positive compartments (Figs. 1A and EV1), consistent with previously published findings (Rensen et al, 2021; Francis et al, 2020). To obtain a more detailed view of CA distribution within nuclear speckles, we performed dual-color STED nanoscopy of infected cells immunostained against SRRM2 and HIV-1 CA. SRRM2 did not appear homogeneously distributed within the speckle region at nanoscopic resolution (< 50 nm; Fig. 1B), but formed distinct and separated subclusters in the range of ~50–100 nm. STED images revealed multiple CA-positive objects clustering in internal speckle regions between these SRRM2 subclusters (Fig. 1B,C). Their size and labeling intensity indicated that these objects may not represent individual capsids, but rather correspond to accumulations of capsids and/or capsid remnants (Fig. 1B,C).

To further elucidate the ultrastructure of these viral objects, we applied CLEM-ET, using IN fused to the self-labeling SNAP-tag and stained with silicone rhodamine (SiR) as a marker to target the positions of nuclear subviral complexes. Infected MDM were subjected to cryo-immobilization by high-pressure freezing, followed by freeze-substitution and plastic embedding. Positions of interest were identified in thin sections based on IN.SNAP.SiR fluorescence using light microscopy and analyzed by ET. Clusters of elongated or conical electron-dense objects with sizes and morphologies consistent with HIV-1 capsids were observed at positions of IN.SNAP.SiR signals (Fig. 1D–G). Some of these capsids appeared morphologically indistinguishable from virion-associated capsids, while broken structures and capsid remnants were also observed. Some structures contained internal density, likely representing the viral nucleic acid complex, while others appeared empty (Fig. 1D–G). These data support recent findings that seemingly intact capsids with active reverse transcription can enter the nucleus of MDM (Kreysing et al, 2025) and reveal their subsequent accumulation and breakage in clusters within the central region of nuclear speckles.

## Nuclear speckle localization of capsids can be perturbed using LEN

Translocation of subviral HIV-1 complexes to nuclear speckles and their retention in the speckle area have been shown to be dependent on CPSF6, which forms a layer on the capsid inside the nucleus (Francis et al, 2020; Li et al, 2020; Selyutina et al, 2020; Rohlfes et al, 2025; Bejarano et al, 2019). Given the overlapping binding site of LEN and CPSF6, we analyzed the influence of LEN or PF74 on the retention of speckle-associated HIV-1 capsids in MDM.

First, we determined the LEN concentration required for the displacement of CPSF6 from nuclear capsids in MDM. To ensure completion of the post-entry phase, drug treatment was initiated at 72 h p.i. MDM infected with IN.eGFP labeled particles were treated with different concentrations of LEN for 1 h, immunostained for CPSF6 or CA, and imaged using quantitative 3D confocal spinning disc microscopy. Full displacement of CPSF6 from nuclear subviral complexes in MDM was observed at a concentration of 500 nM LEN (Fig. 2A,B). Concomitantly, CA epitopes were exposed on these complexes, enabling efficient immunostaining (Fig. 2C,D). The same effect was achieved with 15 μM PF74 in primary MDM (Fig. EV2A–D), similar to results obtained with PF74 in model cell lines and T cells (Müller et al, 2021; Ay et al, 2025). The LEN concentration required to displace CPSF6 from nuclear HIV-1 capsids was 10-fold lower in HeLa-based cells compared to MDM; 5 nM LEN and 50 nM LEN, respectively, resulted in partial and full displacement of CPSF6 and CA epitope exposure in HeLa-based cells (Fig. EV3A–D). LEN binds the capsid hexameric lattice with a $K_D$ of ~ 200 pM (Faysal et al, 2024; Briganti et al, 2025), and the EC50 for inhibition of replication is ~50 pM in several cell lines. However, LEN inhibits different post-entry steps at different concentrations (Bester et al, 2020). Nuclear import of HIV-1 complexes is completely blocked at 5 nM LEN in HeLa-based cells, while reverse transcription requires a concentration of 50 nM for full inhibition (Bester et al, 2020). Hence, the concentration of LEN required to fully displace CPSF6 within one hour of drug treatment of HeLa-based cells is in a similar range as the concentration needed to block reverse transcription, but notably higher than the concentration required to block nuclear import in these cells.

LEN- or PF74-induced removal of CPSF6 from nuclear speckle-associated HIV-1 subviral complexes resulted in their rapid (within 1 h) exit from speckles into the adjacent interchromatin space in MDM (Figs. 2E,F and EV2E,F). This result was similar to the effect of PF74 in model cell lines and T cells, where CPSF6 displacement and exit of HIV-1 complexes were also reported (Francis et al, 2020). However, these authors did not observe rapid relocalization of HIV-1 complexes from nuclear speckles in MDM, while this was clearly observed in the current report. Our results indicate a significant concentration difference for LEN- and PF74-induced effects in MDM compared to other cell types, and this may have obscured the effect in the prior study. To evaluate whether tagged IN signals exiting from nuclear speckles correspond to actively reverse transcribing nuclear HIV-1 complexes, we also analyzed infected MDM labeled with EdU and subsequently treated with 500 nM LEN for IN.eGFP and EdU localization with respect to the speckle marker SRRM2 (Figs. 2E,F and EV2E,F). Following LEN treatment, EdU and IN.eGFP colocalized in the vicinity of nuclear speckles but clearly separated from the SRRM2 signal. We thus conclude that productively reverse-transcribing HIV-1 complexes

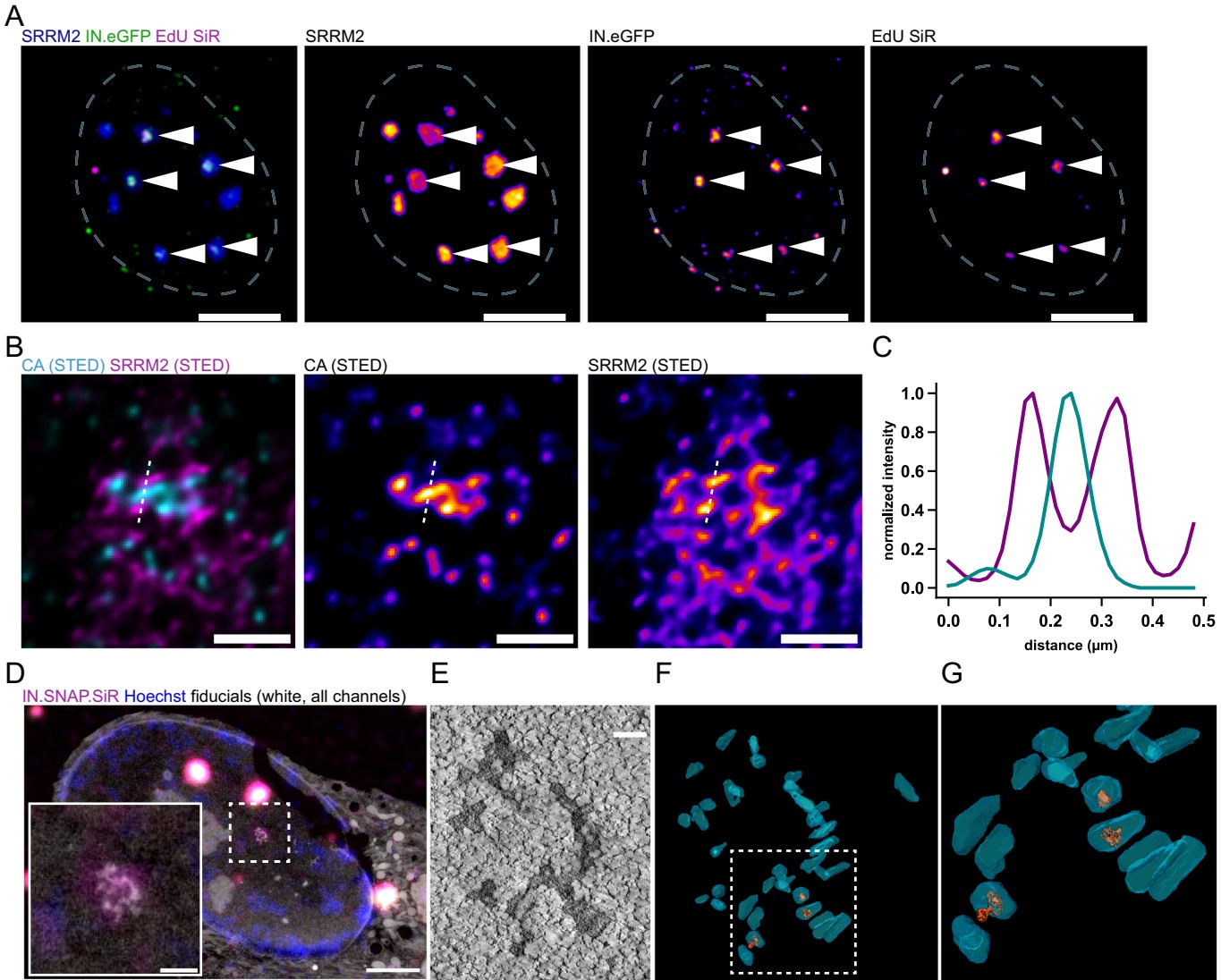

**Figure 1.  Super-resolution and CLEM-ET analyses of capsid clusters in nuclear speckles of MDM.**

(A–C) Super-resolution analyses of HIV-1 cDNA within nuclear speckles of MDM showing EdU IN.eGFP (A) and CA (B) signals in a central region of SRRM2 condensates. One of three independent experiments. (A) Representative images of a maximum intensity projection of an MDM nucleus (white dashed line). Cells were infected for 72 h with VSV-G pseudotyped NNHIV carrying IN.eGFP (green) in the presence of EdU followed by fixation, EdU click labeling (magenta) and immunofluorescence staining using an antibody against SRRM2 (blue). Samples were imaged using Airyscan microscopy. Scale bars: 5 µm. (B) Dual-color STED images of immunostained CA and SRRM2 of MDM infected with VSV-G pseudotyped NNHIV for 72 h. Signal intensities were quantified along the white dotted line and plotted in (C). Scale bars: 0.5 µm. (C) Quantification of signal intensities along the white dotted line in (B) normalized to the highest value. (D–G) CLEM-ET analysis of infected MDM. Cells were cryo-immobilized by high-pressure freezing followed by freeze-substitution and plastic embedding. (D) CLEM overlay (with inverted EM image) of the 250 nm thin section of the cell, positive for IN.SNAP.SiR (magenta), post-stained with Hoechst (blue), and decorated with multifluorescent fiducials for correlation (all channels, white). Scale bars: 2.5 µm (overview) and 500 nm (enlargement). (E) A single slice of the reconstructed electron tomogram correlated to the IN.SNAP.SiR signal boxed in (D). Scale bar: 100 nm. (F, G) 3D rendering of the tomogram shown in (E) and enlargement of the boxed region in (F). See Movie EV1. Source data are available online for this figure.

still encased inside the viral capsid exit from nuclear speckles in MDM upon LEN or PF74-induced CPSF6 removal.

## Functionally complete, integration-competent nuclear HIV-1 dsDNA can be exposed by treatment of infected MDM with CA targeting drugs

We had previously shown that HIV-1 genomic dsDNA separates from the structural and reporter proteins (IN.eGFP) over time

(Müller et al, 2021). For this, we made use of the ANCHOR system, where a GFP-fused DNA-binding protein (OR3) can bind repeats of its cognate binding sequence (ANCH) in the HIV-1 genome, provided that the sequence exists as dsDNA and is accessible to the fusion protein. Since the ANCH repeats were inserted into the viral *env* gene, the system detects only genomes that have been almost completely or completely reverse-transcribed into dsDNA. To detect accessible HIV-1 dsDNA genomes and examine their spatial relationship with, and potential alterations of, the associated capsid

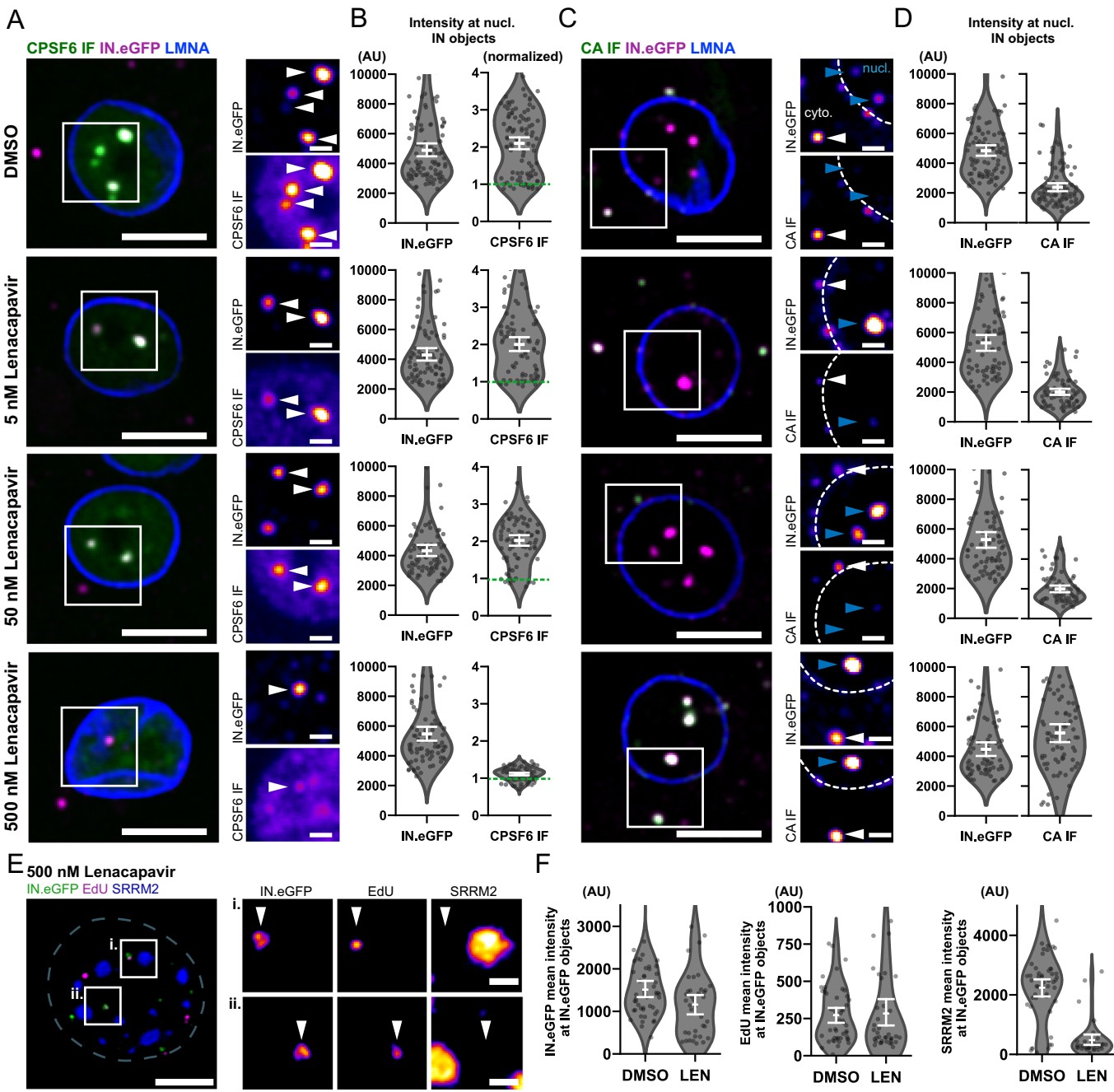

**Figure 2. LEN treatment displaces CPSF6 from nuclear capsids, exposes masked CA epitopes, and leads to the exit of subviral structures from nuclear speckles in MDM.**

(A–F) MDM were infected with IN.eGFP labeled VSV-G pseudotyped NNHIV for 72 h prior to the addition of indicated concentrations of LEN for 1 h. Cells were fixed and immunostained before SDCM (A–D) or Airyscan (E, F) 3D imaging. Samples were stained for CPSF6 (A, B), CA (C, D), LMNA (A–D), or SRRM2 (E). The figure shows maximum intensity projections of the central nuclear region (A–D) or a single z plane (E) of one of three (A–D) or two (E, F) independent experiments. Scale bars: 5 μm (overviews) and 1 μm (enlargements). (A–D) Displacement of pre-assembled CPSF6 (A, B) and exposure of masked CA epitopes (C, D) by LEN. (A) White arrowheads indicate nuclear IN.eGFP objects. (C) White arrowheads indicate cytoplasmic IN.eGFP objects, whereas blue arrowheads indicate nuclear IN.eGFP objects. Dotted lines indicate the nuclear boundary. (B, D) Images were analyzed by automated quantification in 3D using custom-made Python code as described in the Methods and protocols section. CPSF6 signals (B) were normalized to the mean nuclear CPSF6 expression level of the respective cell (green dotted line at $y = 1$). Error bars represent SEM. (E) MDM were infected in the presence of EdU and click labeled prior to immunostaining for SRRM2. IN.eGFP objects displaced from nuclear speckles following treatment with 500 nM LEN for 1 h prior to fixation. One of three independent experiments. (F) Quantification of the experiment described in (E). Mean intensity of IN.eGFP (left panel), EdU (middle panel), and SRRM2 (right panel) at positions of IN.eGFP objects were quantified in samples treated with DMSO or 500 nM LEN for 1 h prior to fixation. Error bars represent 95% CI. Source data are available online for this figure.

structures, we infected a HeLa-derived cell line stably expressing eGFP.OR3 with NNHIV-ANCH also carrying IN.SNAP as a protein marker. At 23 h p.i., infected cells were treated with 15 μM PF74 or DMSO solvent for 1 h and analyzed for the number of detectable eGFP.OR3 signals in the nucleus and for the colocalization of these signals with IN.SNAP and CA. Most IN.SNAP-positive objects did not colocalize with eGFP.OR3 positive objects in DMSO-treated control cells (Fig. 3A), and eGFP.OR3 positive objects were localized outside of nuclear speckles (Fig. EV3E). The number of eGFP.OR3 positive nuclear objects increased two- to threefold in PF74-treated cells compared to control cells (average of 14 instead of 6 objects per nucleus; Fig. 3B,C). Furthermore, the proportion of IN.SNAP/CA signals colocalizing with eGFP.OR3 increased almost threefold from ~25% measured in control cells to ~70% in the PF74-treated sample (Fig. 3B,D, empty arrowheads). Of note, despite the observed increase in eGFP.OR3 signals inside the nucleus, cytoplasmic IN.SNAP-positive objects remained devoid of eGFP.OR3 (Fig. 3B, filled arrowheads), consistent with previous reports that reverse transcription is only completed in the nucleus (Dharan et al, 2020; Burdick et al, 2020; Müller et al, 2021).

Similar results were obtained with LEN (Figs. 3E–H and 4A,B,D–F). Live imaging indicated that eGFP.OR3 signals appeared within minutes after addition of LEN at nuclear diffraction limited IN.SNAP signals (which presumably represent clusters of subviral particles based on the CLEM data described above) (Fig. 3E,F; Movie EV2). Subsequently, we observed movement of IN.SNAP/eGFP.OR3 positive clusters to positions of nearby HIV-1 dsDNA signals (eGFP.OR3), which had already been present prior to the start of imaging (Fig. 3G,H; Movie EV3), consistent with a LEN-induced relocation of IN.SNAP objects out of nuclear speckles and into speckle-associated chromatin domains (compare Figs. 2E and EV2E,F). Both effects together explain the observed enhanced colocalization between eGFP.OR3 and IN.SNAP (Figs. 3B,D and 4A,B,E). The rapid appearance of new OR3 signals upon drug treatment suggested that complete or nearly complete viral cDNA was already present within the respective subviral complexes, but was still shielded by the viral capsid, the CPSF6 coat or another component of the nuclear speckle.

The appearance of new OR3 signals colocalizing with HIV-1 subviral complexes could also be explained if capsid breakage would promote more efficient reverse transcription, or if previously stalled reverse transcription complexes would resume reverse transcription to complete dsDNA upon opening of the capsid. To evaluate these possibilities, the reverse transcriptase inhibitor Efavirenz (EFV) was added 45 min before the addition of LEN to cells infected for 16 h (Fig. 4A–F). This treatment blocks any further reverse transcription immediately before and during subsequent LEN treatment. The average number of mScarlet.OR3 foci per nucleus (Fig. 4D) and the colocalization between IN.eGFP and mScarlet.OR3 were again two- to threefold higher in LEN-treated cells compared to the DMSO control (Fig. 4E), independent of the absence or presence of EFV. The average number of IN.eGFP-positive objects per nucleus was not affected by these treatments (Fig. 4F).

To expand this analysis to MDM and test whether HIV-1 cDNA revealed by LEN from CPSF6-coated nuclear capsids is competent for integration into host cell chromatin, we performed quantitation of integrated HIV-1 proviral sequences in LEN- or control-treated

MDM using Alu-LTR qPCR. Since NNHIV is not competent for integration, we infected primary MDM from three different donors with VSV-G pseudotyped NL4-3 wild-type virus for 70 h before treating the cells with LEN or DMSO for 1 h or 4 h, respectively. To exclude the possibility that a change in the number of integration events after drug treatment could be due to continued nuclear import and reverse transcription, thereby confounding the result, we added EFV to all samples at the time of DMSO or LEN treatment, effectively blocking further reverse transcription. LEN treatment for 1 h or 4 h led to a ca. twofold increase of Alu-LTR qPCR products (Fig. 4G), indicating that integration-competent, functionally complete proviral HIV-1 cDNA could be revealed from CPSF6-coated nuclear capsids by this treatment.

## LEN treatment alters capsid morphology in infected cells

Drug-induced accessibility of pre-synthesized nuclear HIV-1 dsDNA indicated that LEN or PF74 binding may affect capsid structure and integrity besides removing its CPSF6 coat, in accordance with prior in vitro experiments (Faysal et al, 2024). In order to morphologically characterize potential drug-induced changes on nuclear HIV-1 capsids, we employed CLEM-ET. We infected mScarlet.OR3 expressing cells with IN.SNAP.SiR-labeled NNHIV for 24 h and treated the cells with 500 nM LEN, 15 μM PF74, or DMSO for 1 h before cryoimmobilization by high-pressure freezing, freeze-substitution, and plastic embedding. Electron tomograms were recorded at positions correlated to either nuclear IN.SNAP.SiR objects or mScarlet.OR3 objects.

As observed previously (Müller et al, 2021), tomograms correlated to nuclear IN.SNAP-positive positions (independent of additional mScarlet.OR3 signal presence) in control cells consistently revealed clusters of capsid-like objects often containing electron-dense material inside (Fig. 5A, panels i.), whereas tomograms recorded at mScarlet.OR3 positions lacking IN.SNAP did not show a defined structure but rather a density distribution resembling surrounding chromatin (Fig. 5A, panels ii.). In PF74-treated cells, tomograms were recorded at IN.SNAP.SiR positive positions that often were adjacent to, but not directly colocalizing with mScarlet.OR3. We observed structures closely resembling conical HIV-1 capsids that often contained electron-dense material inside. Some of these structures exhibited thin, electron-dense protrusions emanating from the narrow end of the cone-shaped capsid (Fig. 5B, cyan arrows). The same region also revealed broken capsid-like remnants (Fig. 5B). The addition of 500 nM LEN had a similar effect, again showing dense clusters of cone-shaped structures with electron-dense material inside and resembling HIV-1 capsids (Fig. 5C,D). The arrangement of HIV-1 complexes appeared to be more condensed and electron-dense compared to the DMSO control and to previous results from untreated cells (Zila et al, 2021; Müller et al, 2021; Schifferdecker et al, 2022). Given that native capsids isolated from HIV-1 particles aggregate rapidly into dense clusters (Welker et al, 2000), we attribute this observation to the removal of the CPSF6 coat from nuclear HIV-1 capsids. Furthermore, striking electron-dense protrusions were frequently observed, always emanating from the narrow end of cones (Fig. 5C, cyan arrowheads; Fig. 5D, white arrowheads). In addition to these characteristic structures, 3D rendering of tomographic reconstructions revealed abundant broken and apparently connected empty lattices (Fig. 5D, red stars), seemingly oriented with their narrow ends towards clusters of structures (Fig. 5D; Movie EV4).

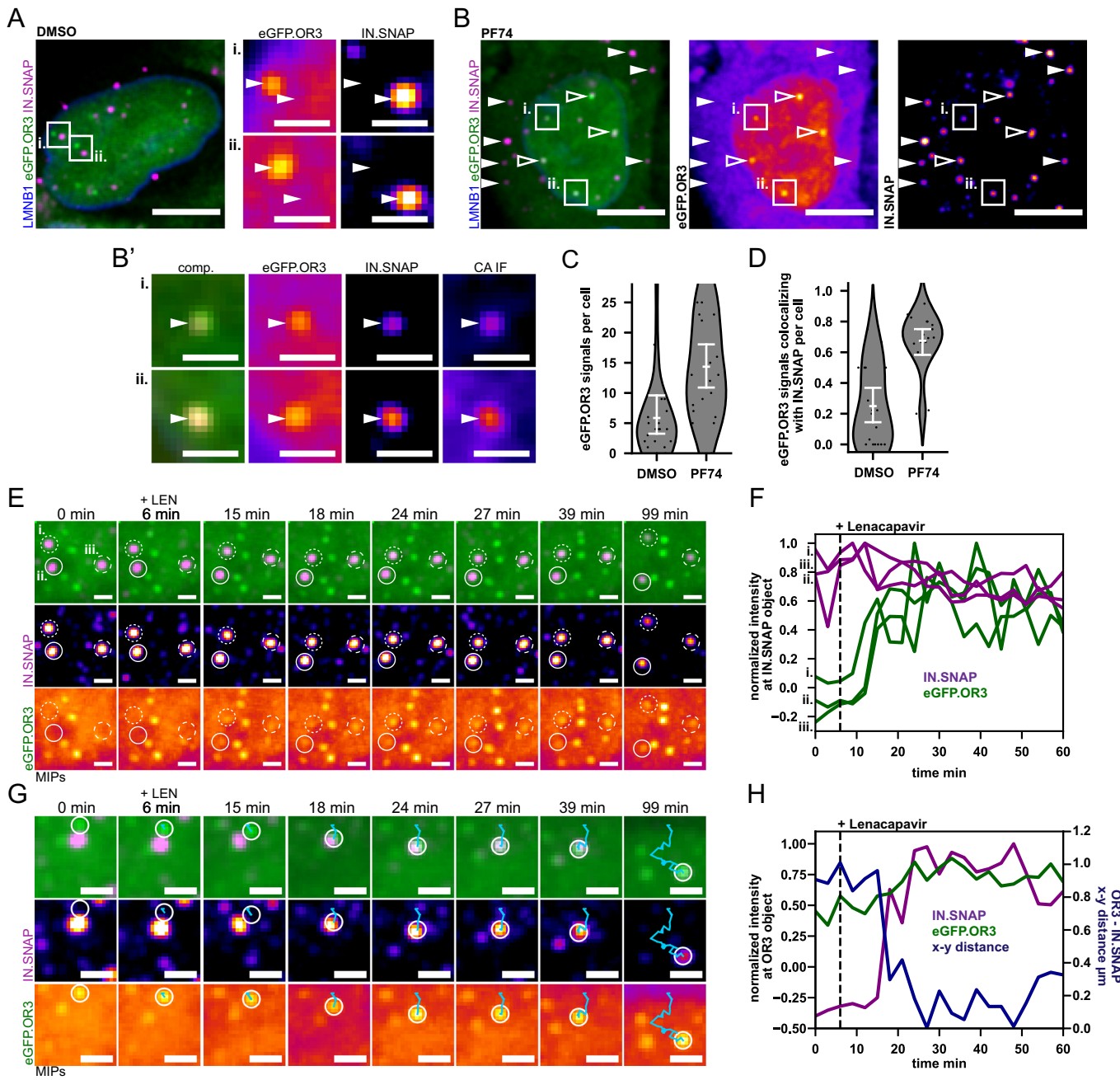

**Figure 3. Exposure of nuclear HIV-1 dsDNA (OR3) by treatment with PF74 and LEN.**

(A–H) HeLa-based cells stably expressing eGFP.OR3 and eBFP2.LMNB1 were infected using VSV-G pseudotyped NNHIV-ANCH IN.SNAP and imaged at 24 h p.i. using 3D confocal spinning disc microscopy. One of three independent experiments. (A) Representative control cell treated with DMSO for 1 h. Arrowheads in enlargements indicate positions of nuclear eGFP.OR3 and IN.SNAP punctae, respectively. Scale bars: 5 µm (overview) and 2 µm (enlargements). (B) Cells infected as in (A) and treated with 15 µM PF74 for 1 h before fixation. Arrowheads indicate IN.SNAP punctae in the cytosol (filled) or nucleus (open), respectively. Note that no eGFP.OR3 punctae are detected on IN.SNAP-positive objects in the cytosol. Scale bars: 5 µm. (B') Enlargements show two exemplary complexes boxed in the nuclear region of (B). Scale bars: 2 µm. (C, D) Quantification of the total number of nuclear eGFP.OR3 objects per cell (C) and the fraction of eGFP.OR3 signals colocalizing with IN.SNAP objects per cell (D) in DMSO and 15 µM PF74-treated cells. (E–H) Live imaging of cells using 3D confocal spinning disc microscopy. New eGFP.OR3 signals appear at positions of IN.SNAP objects (E, F) and IN.SNAP objects move to positions of previously present eGFP.OR3 objects (G, H). Recording starts at 22 h p. i. with a time resolution of 3 min per frame. LEN (500 nM) was added 6 min after the start of imaging. (E, G) Maximum intensity projections (MIPs) of representative nuclear events. The circles indicate tracked IN.SNAP (E, objects i.-iii.) or eGFP.OR3 (G) objects. See Movie EV2 (E, F) and Movie EV3 (G, H). Scale bars: 2 µm. (F) Quantification of signal intensities of the three IN objects (i.-iii.) circled in (E). (H) Quantification of signal intensities of the OR3 object circled in (G). Distance to the centroid of the IN.SNAP object is presented on the right y axis. (F, H) Plotted are mean intensities (diameter 440 nm) normalized with the local background mean intensities (ring diameter 440–1100 nm from centroid) at IN.SNAP (F) or eGFP.OR3 (H) objects in both, the IN.SNAP and the OR3 channels. Source data are available online for this figure.

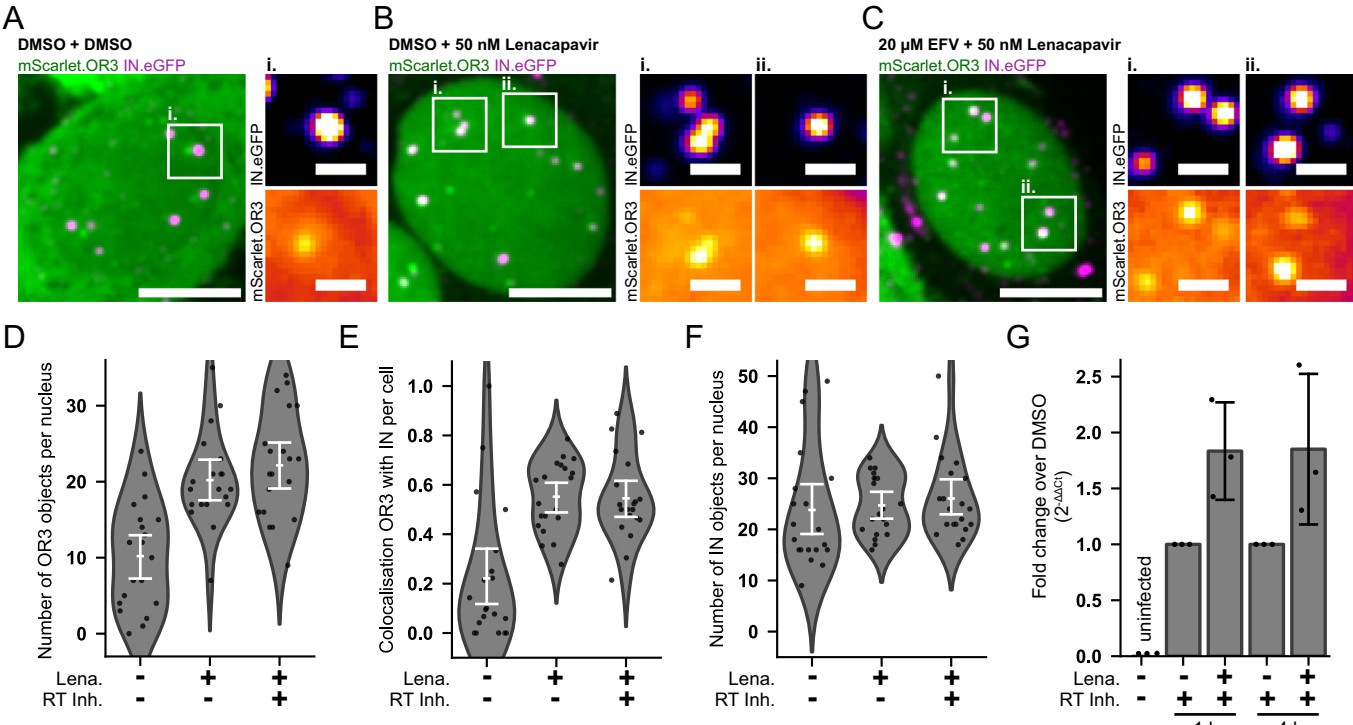

**Figure 4. Exposure of additional OR3 signals in the nucleus by LEN treatment occurs independently of ongoing reverse transcription and leads to increased genomic HIV-1 integration.**

(A–F) mScarlet.OR3 expressing TZM-bl cells were infected with VSV-G pseudotyped IN.eGFP labeled NNHIV for 16 h prior to addition of the RT inhibitor EFV or DMSO vehicle and incubated for 45 min, after which 50 nM LEN was added or not for 1 h. One of three independent experiments. (A–C) Representative examples of cells treated with DMSO/DMSO (A), DMSO/50 nM LEN (B), and 20 μM EFV/50 nM LEN (C). Scale bars: 5 μm (overviews) and 1 μm (enlargements). (D–F) Quantification of the number of mScarlet.OR3 objects per nucleus (D), the fraction of mScarlet.OR3 signals colocalizing with IN.eGFP per nucleus (E), and the number of IN.eGFP objects per nucleus (F). Shown is the mean, and the error bars represent 95% CI. Points represent single cells. (G) Quantitation of integrated HIV-1 genomic DNA. MDM were infected with VSV-G pseudotyped NL4-3 for 70 h prior to the addition of the RT inhibitor EFV together with 500 nM LEN or DMSO and incubated for 1 h or 4 h before isolation of genomic DNA and Alu-LTR qPCR. Data points show qPCR results normalized to the respective DMSO control for three independent blood donors. Shown is the mean and error bars represent standard deviation. Source data are available online for this figure.

# Discussion

While many recent studies strongly enhanced our understanding of the early phase of HIV-1 replication, the processes of intranuclear trafficking, genome uncoating, and how they relate to integration are still incompletely understood. Here, we showed that morphologically intact HIV-1 capsids with interior nucleic acid density as well as broken and empty capsid-like particles clustered in nuclear speckles of post-mitotic primary MDM in a CPSF6-dependent way. Similar clustering was previously observed in various tissue culture-adapted cell lines (Müller et al, 2021; Schifferdecker et al, 2022; Ay et al, 2025). Treatment of cells containing nuclear HIV-1 complexes with capsid-targeting drugs led to CPSF6 removal, exit from nuclear speckles and accessibility of pre-synthesized, integration-competent HIV-1 dsDNA, indicating genome uncoating (Fig. EV4). The observation of broken capsids without interior density is consistent with prior results in different cell lines that also reported incomplete capsid lattice remnants inside the nucleus of HIV-1-infected cells (Müller et al, 2021; Ay et al, 2025). These results indicated physical breakage of the capsid as a mechanism of HIV-1 genome uncoating rather than disassembly of the CA lattice. Other reports suggested rapid disassembly of the capsid lattice (Burdick et al, 2020; Li et al, 2021)

based on loss of fluorescence from nuclear capsids carrying substoichiometric amounts of a GFP-tagged CA fusion protein; further studies will be needed to resolve this difference.

HIV-1 capsids associated with CPSF6, clustered within the center of nuclear speckles in MDM, and contained de novo-synthesized viral DNA. Super-resolution STED analysis of the major nuclear speckle component SRRM2 indicated distinct speckle subdomains, consistent with previous findings from HeLa, HEK293T, and WI-38 (human lung fibroblast) cells obtained using structured illumination microscopy (SIM) (Fei et al, 2017; Zhang et al, 2024) and stochastic optical reconstruction microscopy (STORM) (Zhang et al, 2024). SRRM2 and the second major nuclear speckle component SON inhabit distinct immiscible phases in nuclear speckles, each enriched with different speckle components (Zhang et al, 2024). These substructures likely represent clusters of ~40 nm ordered assemblies driven by microphase separation of block copolymer-like splicing factors (preprint: Shinn et al, 2025). Interestingly, we observed that capsid clusters localized to the subdomains lacking SRRM2. We hypothesize that not only the immersion of capsids into the phase-separated speckle environment (Xu et al, 2022), but also this specific subdomain localization is driven by the biophysical properties of the multimeric CPSF6 coat (Wei et al, 2022) encasing nuclear capsids. Our live

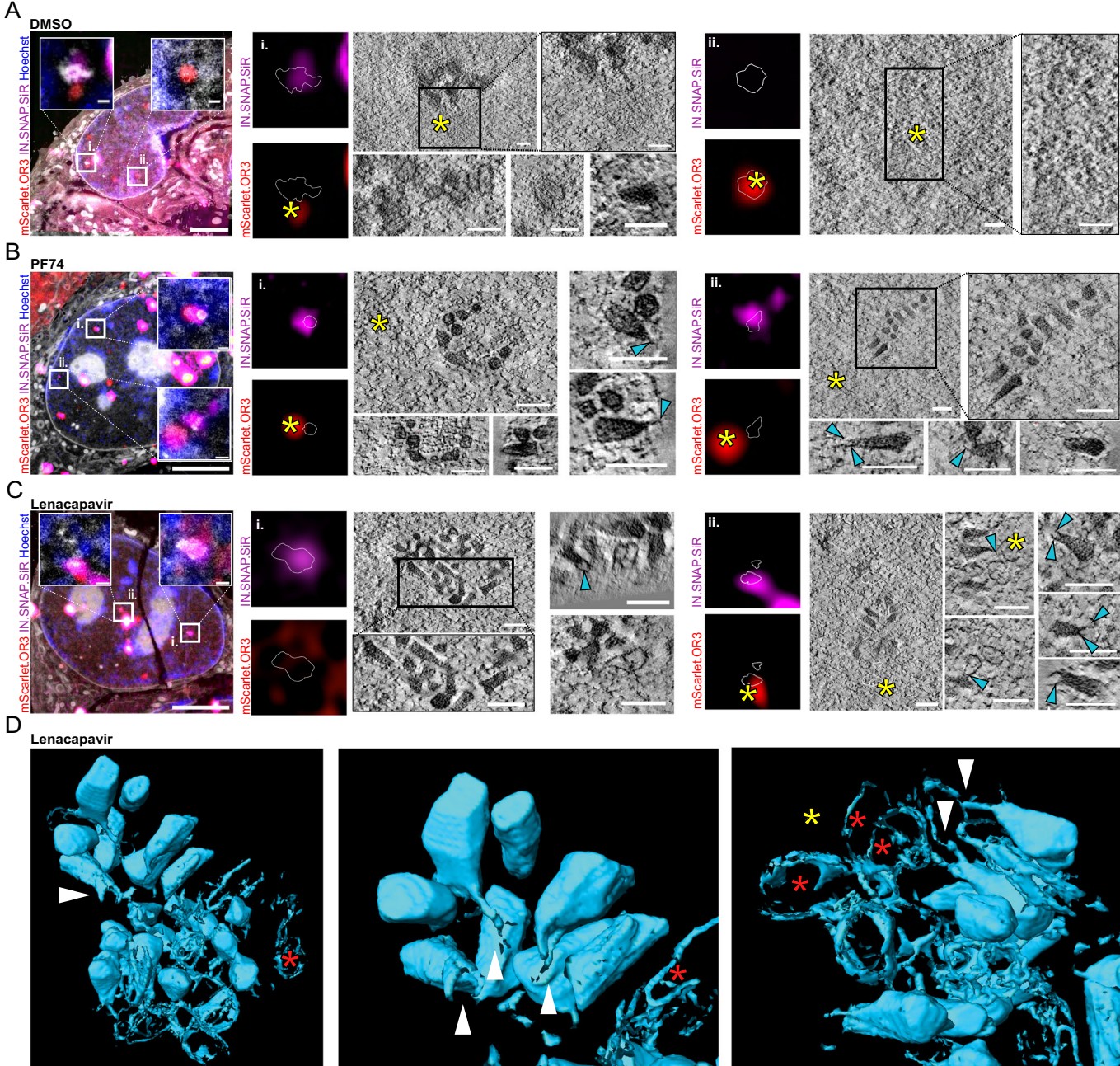

**Figure 5. Correlative light and electron microscopy with electron tomography (CLEM-ET) analyses of capsid alterations following LEN/PF74 treatment.**

(A–D) mScarlet.OR3 expressing HeLa-based TZM-bl cells infected with VSV-G pseudotyped IN.SNAP.SiR-labeled NNHIV were treated at 24 h p.i. with DMSO (A), 15 μM PF74 (B), or 500 nM LEN (C, D) for 1 h prior to cryoimmobilization, freeze-substitution, and further processing for CLEM. Electron tomograms were acquired at positions correlated to IN.SNAP.SiR and/or mScarlet.OR3 signals. (A–C) Left panels show CLEM overlays of the 250 nm thin sections with the inverted EM image, positive for mScarlet.OR3 (red) and/or IN.SNAP.SiR (magenta), post-stained with Hoechst (blue), and decorated with multifluorescent fiducials for correlation (all channels, white). Slices of individual tomograms correlated to the indicated positions are presented. Cyan arrowheads indicate protrusions from the narrow ends of conical capsid objects. Scale bars: 2.5 μm (overlay) and 100 nm (tomograms). (A) Tomograms of DMSO-treated cells correlated with an IN.SNAP.SiR signal with an mScarlet.OR3 signal in close proximity (yellow star, position i.) and correlated to an mScarlet.OR3 signal (yellow star) lacking IN.SNAP.SiR (position ii.). (B) PF74-treated cells correlated with IN.SNAP.SiR signals associated with mScarlet.OR3 signals (yellow stars, both positions). (C) LEN-treated cells with presented tomograms correlated to IN.SNAP.SiR positions lacking mScarlet.OR3 (position i.) or overlapping with mScarlet.OR3 signal (yellow star, position ii.). (D) 3D rendering of segmented electron densities of tomogram shown in (C, position ii.). Red stars indicate empty/fused lattices, and white arrowheads point to protrusions at the narrow ends of conical capsid objects. See Movie EV4. Source data are available online for this figure.

imaging and CLEM data showed that removing the CPSF6-layer by LEN or PF74 treatment forced the clustered capsids to move out of nuclear speckles in a concerted fashion, remaining densely clustered in the adjacent nucleoplasm. Of note, these structures often followed the path of previously uncoated viral dsDNA, which had dissociated from the viral structures inside nuclear speckles before drug treatment. While this may simply reflect the 3D chromatin architecture constraining possible trajectories, it is also conceivable that specific interactions between the exposed dsDNA and the remaining subviral complex influence these dynamics.

Treatment of infected cells carrying nuclear HIV-1 complexes with LEN or PF74 induced exposure of full-length or nearly full-length viral dsDNA to be recognized by a fluorescent 66 kDa fusion protein binding a specific DNA sequence in the dsDNA genome. This effect was independent of ongoing reverse transcription, indicating that complete or almost complete dsDNA of HIV-1 can be synthesized within an apparently intact HIV-1 capsid that excludes access of the fusion protein. Furthermore, LEN treatment also induced a twofold increase of integrated HIV-1 genomes when primary MDM were infected by wild-type virus. This effect did not require ongoing reverse transcription, indicating that functionally complete and integration-competent genomic HIV-1 DNA had been synthe-sized inside nuclear capsids and could be revealed by LEN treatment. These observations are consistent with the ability of HIV-1 to evade activation of innate immune responses (Cingöz and Goff, 2019) unless cGAS-mediated sensing is induced by PF74 (Sumner et al, 2020; Ay et al, 2025) or LEN (Scott et al, 2025).

Our findings are consistent with reports suggesting that synthesis of long dsDNA inside the HIV-1 capsid is important for capsid breakage and genome uncoating, but suggest that reverse transcription may not be sufficient. ANCHOR detection in our system requires the dsDNA to be completed to at least 7.3 kbp. This DNA length is well above the critical limit of ~ 6 kbp dsDNA for efficient HIV-1 genome uncoating reported in a previous study (Burdick et al, 2024). Furthermore, the observed increase of integration events as shown by Alu-LTR qPCR and independent of ongoing reverse transcription shows that at least some of the nuclear capsids contain functionally complete viral genomes that were not accessible prior to LEN treatment.

We hypothesize, therefore, that additional factors besides long dsDNA synthesis are important to reveal the HIV-1 genome to the nuclear environment. Given that capsid-targeting drugs remove the CPSF6 coat and induce capsid breakage, we cannot currently distinguish which of these mechanisms is most relevant for the observed phenotype. Drug-induced displacement of the CPSF6 coat could make the capsid more sensitive to breakage by dsDNA or could expose a previously broken capsid lattice that had been stabilized by the outer CPSF6 coat. Alternatively, genome exposure may depend on the direct effect of PF74 or LEN binding on capsid lattice stability that has been demonstrated in vitro (Faysal et al, 2024). Conceivably, capsid integrity may become compromised already upon capsid passage through the narrow NPC channel. This was not apparent in a recent cryo-electron tomography study, but complete tracing of the lattice of NPC-associated and nuclear HIV-1 capsids was not possible due to the low number of events and dense surrounding environment (Kreysing et al, 2025). In the described scenario, CPSF6 coating of the capsid as it exits the NPC channel would shield and stabilize the subviral structure, and removal of CPSF6 may then be sufficient for

breaking the lattice by the forces of internal dsDNA. We can currently not distinguish which of these effects, or a combination thereof, are critical for drug-induced DNA exposure in our experiments. Independent of the precise mechanism, our results together with those of Burdick et al (Burdick et al, 2024), indicate that synthesis of long dsDNA and factor(s) affecting the integrity of the subviral structure in nuclear speckles may act together to induce efficient capsid opening and genome uncoating and may also ensure that this occurs at the optimal position for genome integration into transcriptionally active chromatin. This may not be an all-or-none phenomenon, but rather influence the kinetics and efficiency of uncoating. Furthermore, these processes may be affected by the availability of host factors in different cell types, consistent with the observation of different LEN concentrations required for CPSF6 removal and DNA exposure depending on the cell type.

LEN treatment resulted in a clear change of HIV-1 capsid morphology within the nucleus, frequently inducing bifurcated linear protrusions emanating from the narrow ends of capsid cones. These alterations were detected at LEN concentrations that remove the CPSF6-layer from the hydrophobic binding pocket, and which substantially exceed antiviral EC50 values of the inhibitor. Antiviral LEN activity at lower concentrations could previously be attributed to binding of LEN to unoccupied hydrophobic pockets without displacing cellular cofactors (Wei et al, 2022). The protrusions could represent structurally altered parts of the capsid lattice, and/ or the (chromatinized) viral dsDNA complex emanating from an opening in the capsid. We did not observe direct colocalization of the HIV-1 dsDNA-binding fusion protein with IN.SNAP.SiR positive positions, but this may have been due to fluorescence-intensity losses by CLEM sample preparation, while brighter mScarlet.OR3 signals that existed prior to drug treatment were retained. Clusters of capsids displaying electron-dense protrusions at their narrow ends appeared in close proximity to networks of empty and flattened lattice structures possibly connected to each other, consistent with the observation that LEN and related compounds both disrupt the integrity of the mature HIV-1 capsid and at the same time hyperstabilize the hexameric CA lattice (Faysal et al, 2024; Li et al, 2025). Given that LEN binding is incompatible with CA pentamers (Huang et al, 2025), we speculate that local conversion of pentamers may lead to disruption and opening of the capsid structure. Pentamer density is highest at the narrow end of the cone, which can explain the preferential deformation and opening of capsids in this region, allowing release and chromatinization of the viral genome from the capsid interior upon drug treatment. Drug-induced stabilization of the remaining hexameric capsid lattice can also explain the high prevalence of flattened or apparently fused lattice remnants in our study. Mechanistically, this might be achieved by binding of PF74 or LEN to the Thr-Val-Gly-Gly motif within CA, a region which mediates a structural switch between a pentamer- and hexamer-favoring conformation (Schirra et al, 2023; Stacey et al, 2023). The altered capsid morphology we observed in the nuclei of infected cells is consistent with recent negative-stain electron micrographs of isolated capsids treated with LEN in vitro (Li et al, 2025). However, the drug-induced phenotype does not necessarily reflect uncoating in the absence of drugs, and endogenous reverse transcription in isolated native HIV-1 capsids in vitro induced lattice breakage and polynucleotide loop emanation at the broad end and long side of the capsid (Christensen et al, 2020).

In summary, our results point to additional factor(s) involved in capsid breakage and genome release besides the synthesis of a sufficiently long dsDNA contributing the physical force. These may include factors destabilizing the CPSF6 coat from speckle-associated capsids and/or factors destabilizing the capsid lattice. Once the capsid is broken, rapid chromatinization of exposed dsDNA loops (Christensen et al, 2020) via histones (Geis and Goff, 2019; Wang et al, 2016) or binding of other proteins to exposed DNA loops may facilitate genome exit from broken capsid remnants. Understanding the detailed mechanisms and factors involved in nuclear HIV-1 genome uncoating and subsequent integration can also help to better understand latency induction and maintenance, with the aim to devise efficient strategies to purge the latent reservoir in infected individuals. Our results also shed further light on the mechanism(s) of action of the potent anti-HIV drug LEN.

# Methods

### Reagents and tools table

| Reagent/resource | Reference or source | Identifier or catalog number |
|---|---|---|
| **Experimental models** | | |
| TZM-bl | Wei et al, 2002; https://doi.org/10.1128/AAC.46.6.1896-1905.2002 | RRID:CVCL_B478 |
| HEK293T | Pear et al, 1993; https://doi.org/10.1073/pnas.90.18.8392 | RRID:CVCL_0063 |
| TZM-bl mScarlet.OR3 IRES puro | Müller et al, 2021; https://doi.org/10.7554/eLife.64776 | – |
| TZM-bl eBFP2.LMNB1 IRES BLR eGFP.OR3 IRES puro | Müller et al, 2021; https://doi.org/10.7554/eLife.64776 | – |
| MDM derived from buffy coats of healthy human blood donors | – | – |
| **Recombinant DNA** | | |
| pNNHIV env(stop) ANCH | Müller et al, 2021; https://doi.org/10.7554/eLife.64776 | – |
| pVpr-IN$_{D64N/D116N}$.SNAP | Müller et al, 2021; https://doi.org/10.7554/eLife.64776 | – |
| pVpr-IN$_{D64N/D116N}$.eGFP | Müller et al, 2021; https://doi.org/10.7554/eLife.64776 | – |
| pCMV-VSV-G | Expression of VSV-G; B. Weinberg (Whitehead Institute, MA, USA) | RRID:Addgene_8454 |
| pNLC4-3 | Bohne et al, 2004; https://doi.org/10.1016/S0014-5793(04)00277-7 | – |
| **Antibodies** | | |
| Mouse anti-hLamin A/C | Santa Cruz Biotech | Cat#:sc-7292; RRID:AB_627875 |

| Reagent/resource | Reference or source | Identifier or catalog number |
|---|---|---|
| Rabbit anti-hCPSF6 | Atlas Antibodies | Cat#:HPA039973; RRID:AB_10795242 |
| Rabbit anti-CA antiserum | Welker et al, 2000, https://doi.org/10.1128/jvi.74.3.1168-1177.2000 | – |
| Rabbit anti-SON antibody | Sigma-Aldrich | Cat#: HPA023535 |
| Mouse anti-SRRM2 (SC35) | Abcam | Cat#: ab11826 |
| Goat anti-mouse Alexa Fluor 405 | Thermo Fisher Scientific | Cat#: A-31553 |
| Goat anti-Rabbit Alexa Fluor 488 | Thermo Fisher Scientific | Cat#: A-11008 |
| Goat anti-Rabbit Alexa Fluor 568 | Thermo Fisher Scientific | Cat#: A-11011 |
| Goat anti-Rabbit Alexa Fluor 647 | Thermo Fisher Scientific | Cat#: A-21245 |
| Goat anti-mouse STAR RED | Abberior | Cat#: STRED-1001-500UG |
| Goat anti-rabbit Atto 594 | Sigma-Aldrich | Cat#: 77671 |
| **Oligonucleotides and other sequence-based reagents** | | |
| Alu1 primer | Tan et al, 2006, https://doi.org/10.1128/jvi.80.4.1939-1948.2006 | TCCCAGCTACTGGGGAGGCTGAGG |
| LM667 primer | Tan et al, 2006, https://doi.org/10.1128/jvi.80.4.1939-1948.2006 | ATGCCACGTAAGCGAAACTCTGGCTAACTAGGGAACCCACTG |
| λT primer | Tan et al, 2006, https://doi.org/10.1128/jvi.80.4.1939-1948.2006 | ATGCCACGTAAGCGAAACT |
| LR primer | Tan et al, 2006, https://doi.org/10.1128/jvi.80.4.1939-1948.2006 | TCCACACTGACTAAAAGGGTCTGA |
| Taqman ZXF-P | Tan et al, 2006, https://doi.org/10.1128/jvi.80.4.1939-1948.2006 | 6-FAM- TGTGACTCTGGTAACTAGAGATCCCTCAGACCC-MGB |
| B13 forward primer | – | CCCCAGGGAGTAGGTTGTGA |
| B13 reverse primer | – | TGTTATTTGAGAAAAGCCCAAAGAC |
| Taqman B13 | – | 6-FAM- CAGCAGGAAAGGAC-MGB |
| **Chemicals, enzymes, and other reagents** | | |
| PF-3450074 (PF74) | Sigma-Alldrich | Cat#: SML0835 |
| Lenacapavir (GS-6207) | Mamuka Kvaratskhelia, University of Colorado | – |
| Efavirenz (EFV) | Sigma-Alldrich | Cat#: SML0536 |
| SNAP-Cell® 647-SiR | New England Biolabs | Cat#: S9102S |
| Paraformaldehyde (PFA) | Thermo Fisher Scientific | Cat#: 043368.9 M |
| EdU (5-ethynyl-2'-deoxyuridine) | Thermo Fisher Scientific | Cat#: E10187 |

| Reagent/resource | Reference or source | Identifier or catalog number |
|---|---|---|
| Albumin Fraction V, fatty acid-free (BSA) | Carl Roth | Cat#: 0052.3 |
| **Software** | | |
| Imspector software v16.1.6905 | Abberior Instruments GmbH | RRID:SCR_015249 |
| ZEN Blue v3.1 | Carl Zeiss Microscopy | RRID:SCR_013672 |
| Fiji/ImageJ v1.53 c | Schindelin et al, 2012; https://doi.org/10.1038/nmeth.2019 | RRID:SCR_002285 |
| Ilastik v1.3.3 | Berg et al, 2019; https://doi.org/10.1038/s41592-019-0582-9 | RRID:SCR_015246 |
| Icy v2.0.3.0 | de Chaumont et al, 2012; https://doi.org/10.1038/nmeth.2075 | RRID:SCR_010587 |
| Seaborn v0.13.0 | Waskom et al, 2021; https://doi.org/10.5281/zenodo.3629446 | RRID:SCR_018132 |
| Volocity v6.3 | Perkin Elmer | RRID:SCR_002668 |
| Prism v5.01 | GraphPad Software Inc. | RRID:SCR_002798 |
| Matplotlib v3.1.3 | Hunter, 2007; https://doi.org/10.1109/MCSE.2007.55 | RRID:SCR_008624 |
| IMOD v4.9.4 | Kremer et al, 1996; https://doi.org/10.1006/jsbi.1996.0013 | RRID:SCR_003297 |
| SerialEM v3.7.9 | Mastronarde, 2005; https://doi.org/10.1016/j.jsb.2005.07.007 | – |
| Amira-Avizo Software v2019.3 | Thermo Fisher Scientific | RRID:SCR_007353 |
| PyMol v1.3 | Schrodinger LLC | RRID:SCR_000305 |
| Python v3.8.5 | – | |
| **Other** | | |
| Click-iT EdU Alexa Fluor 647 Imaging kit | Thermo Fisher Scientific | Cat#: C10340 |
| Quick-DNA Miniprep | Zymo Research | Cat#: D3025 |

## Methods and protocols

### Cell culture

HeLa-based TZM-bl cells (Wei et al, 2002) and human embryonic kidney 293T cells (HEK293T) (Pear et al, 1993) were cultured in Dulbecco's Modified Eagle's Medium (DMEM) supplemented with 10% fetal bovine serum (FBS), 100 U ml$^{-1}$ penicillin, and 100 mg ml$^{-1}$ streptomycin (PAN Biotech, Germany). eGFP.O3, mScarlet.OR3 and eBFP2.LMNB1-expressing TZM-bl cells have been described previously (Müller et al, 2021). Cell lines were tested for mycoplasma contamination (MycoAlert mycoplasma detection kit, Lonza Rockland, USA) and authenticated by STR profiling (Eurofins Genomics, Germany). MDM were isolated from buffy coats of healthy human blood donors and cultured in RPMI 1640 (Thermo Fisher Scientific) containing 10% heat-inactivated FBS, 100 U ml$^{-1}$ penicillin, 100 mg ml$^{-1}$ streptomycin and 5%

human AB serum (Sigma-Aldrich) as described previously (Bejarano et al, 2019).

### Virus stock production

293T cells were seeded into nine T175 dishes (per ultracentrifuge rotor) to reach 50–70% confluency the next day. Cells were then transfected with proviral plasmid NNHIV env(stop) ANCH (Müller et al, 2021), pVpr-IN$_{D64N/D116N}$.SNAP or pVpr-IN$_{D64N/D116N}$.eGFP expression plasmid (Müller et al, 2021), and pCMV-VSV-G (Addgene plasmid #8454, a gift from Bob Weinberg) at a ratio of 7.7:1.3:1.0 µg using calcium phosphate (70 µg total DNA per T175 dish). For the production of integration-competent wild-type virus, the proviral plasmid pNLC4-3 was transfected together with pCMV-VSV-G at a ratio of 7.7:1.0 µg DNA. Medium was changed after 4–8 h, and cells were incubated at 37 °C for 2 days prior to harvesting the supernatant. Centrifugation at $300 \times g$ was performed for 5 min before filtration through 0.45-µm mixed cellulose ester (MCE) filters. Filtered supernatant was overlayed onto 20% (w/w) sucrose and centrifuged at $107,000 \times g$ for 2 h at 4 °C. Viral particles were resuspended in 75 µl PBS containing 10% FCS and 10 mM HEPES (pH 7.5), pooled, and frozen at −80 °C. Viral stocks were quantified using the SYBR Green-based Product Enhanced Reverse Transcription assay (SG-PERT) as described previously (Pizzato et al, 2009).

### Virus labeling, infection of cells, and drug treatments for imaging

$3.33 \times 10^3$ TZM-bl cells or $5 \times 10^3$ MDM per well were seeded into 15-well µ-Slides (Ibidi, Germany) in 50 µl medium and incubated overnight at 37 °C and 5% CO$_2$. BG-SiR (SNAP-SiR, New England Biolabs) was prediluted to 20 µM in medium and added 1:10 to concentrated particles for a final concentration of 2 µM and incubated for 30 min at 37 °C. Cells were infected using 30 µU (TZM-bl) or 60 µU (MDM) RT activity of IN.eGFP or IN.SNAP labeled VSV-G pseudotyped NNHIV env(stop) ANCH. Cells were drug-treated and fixed at the indicated timepoints. 20 µM Efavirenz (EFV, Sigma-Alldrich) was added 45 min prior to indicated concentrations of Lenacapavir (LEN or GS-6207, provided by Mamuka Kvaratskhelia, University of Colorado) or 15 µM PF74 (Sigma-Alldrich). LEN and PF74 were added 1 h prior to washing the cells with PBS and fixation with 4% paraformaldehyde (PFA). EdU (Thermo Fisher Scientific) was added at the time of infection at 10 µM.

### Immunofluorescence staining and EdU click labeling

Fixed cells were permeabilized by 0.5% Triton X-100 for 10 min, followed by washing three times with 3% bovine serum albumin (BSA) in PBS and blocking for 1 h at room temperature. In the case of EdU click labeling, cells were treated with Click-iT EdU Alexa Fluor 647 Imaging kit (Thermo Fisher Scientific, USA) according to the manufacturer's instructions. Primary antibodies (Table 1) were diluted in 0.5% BSA in PBS and incubated for 1 h at room temperature. After three washing steps with 3% BSA in PBS, secondary antibodies (anti-rabbit or anti-mouse goat polyclonal labeled with Alexa Fluor 405, 488, 568, or 647 (Thermo Fisher Scientific) for confocal and Airyscan microscopy, or labeled with STAR RED (Abberior) or Atto 594 (Sigma-Aldrich) for STED microscopy) were diluted 1:1000 (Alexa Fluor labeled antibodies) or 1:500 (STAR RED and Atto 594 labeled antibodies) with 0.5% BSA in PBS and added to the cells. After 1 h incubation at room temperature in the dark, cells were washed three times with PBS and stored at 4 °C in the dark until imaging.

**Table 1.** Primary antibodies.

| Target | Source | Supplier | Cat no. or reference | Dilution |
|---|---|---|---|---|
| Anti-hLamin A/C | Mouse monoclonal | Santa Cruz Biotech. | Cat#:sc-7292 | 1:100 |
| Anti-hCPSF6 | Rabbit polyclonal | Atlas Antibodies | Cat#:HPA039973 | 1:500 |
| Anti-CA antiserum | Rabbit polyclonal | In-house | Welker et al 2000 | 1:1000 |
| Anti-SON antibody | Rabbit polyclonal | Sigma-Aldrich | Cat#: HPA023535 | 1:800 |
| Anti-SRRM2 (SC35) | Mouse monoclonal | Abcam | Cat#: ab11826 | 1:500 |

### Imaging

Confocal spinning disc imaging was performed using an inverted Perkin Elmer Ultra VIEW VoX 3D spinning disk confocal microscope (Perkin Elmer, United States) with a 60× oil immersion objective (NA 1.49; Perkin Elmer) and a pixel size of 0.22 μm. 3D stacks of 10–30 randomly chosen positions were automatically recorded with a z-spacing of 0.5 μm. For live-cell imaging, medium was changed to imaging medium containing FluoroBrite DMEM (Thermo Fisher Scientific), 10% FCS, 4 mM GlutaMAX (Gibco Life Technologies), 2 mM sodium pyruvate (Gibco Life Technologies), 20 mM HEPES pH 7.4, 100 U ml$^{-1}$ Penicillin and 100 μg ml$^{-1}$ Streptomycin (PAN-Biotech), and cells were imaged inside a humid incubation chamber set to 37 °C and 5% $CO_2$ with a time interval of 3 min per stack.

STED imaging was performed using a STED system (Abberior Instruments GmbH, Germany) with a 775 nm depletion laser and 100× oil immersion objective (NA 1.4; Olympus UPlanSApo). Images were acquired in the 590 and 640 laser lines with a nominal STED power set to 80% of the maximal power of 3 W, 20 μs pixel dwell time, and 15 nm pixel size. STED images were deconvolved using the Richardson-Lucy algorithm within the Imspector software (Abberior Instruments GmbH).

Airyscan point laser scanning confocal microscopy was performed on a Zeiss LSM900 microscope (Carl Zeiss Microscopy, Germany) equipped with the Airyscan detector, using an oil-immersion Plan-Apochromat 63× objective (NA 1.4/Zeiss) in the Airyscan super-resolution mode. Multichannel images were acquired sequentially in the stack scanning mode using 405, 488, and 640 nm diode lasers for fluorophores in the blue, green, and far-red spectra, respectively. Emission detection was configured using variable dichroic mirrors to be 400–490 for blue fluorophore detection, 490–580 for green fluorophore detection, and 620–700 for far red fluorophore detection. Airyscan detectors were used with the gain adjusted between 700 and 900 V; offset was not adjusted (0%). Sampling was system-optimized for Airyscan super-resolution imaging (approx. 50 nm in *xy* axis and 130 nm in *z* axis), acquisition was performed bidirectionally with a pixel dwell time between 0.7 and 1.2 μs. Subsequently, ZEN Blue 3.1 software was used for 3D Airyscan processing with automatically determined default Airyscan Filtering (AF) strength.

### Image analyses and data visualization

Presentation of images. For visualization purposes, confocal and Airyscan images were filtered using a 1 pixel mean filter using Fiji/ImageJ (Schindelin et al, 2012) to reduce noise. The standard 'Fire' lookup table within Fiji was used for presenting single-channel images in figures. Linear unmixing was performed in the case of fluorophore crosstalk.

Quantification of images. An automated processing pipeline was established using a combination of in-house Python v3.11.5 code and Ilastik v1.3.3 (Berg et al, 2019). A pixel and object classifier was trained using Ilastik on raw 3D stacks of images to segment the Lamin signal and IN.SNAP spots. Object identities were exported and loaded into a Python script that segmented the nucleoplasm by mathematically filling the nuclear interior and eroding the mask to exclude the nuclear envelope. IN.SNAP objects were then evaluated based on their pixel location. Only objects with all pixels falling within the nucleoplasmic mask were labeled as nuclear. Mean pixel intensities of these nuclear objects were then collected and plotted. CPSF6 signals were normalized to the mean background expression level of the entire nucleoplasm. For quantification of OR3 signals per nucleus, a semi-automated approach using Icy (de Chaumont et al, 2012) was performed. Using the spot detector of Icy, OR3, and IN.SNAP objects of single nuclei were segmented in 3D. All detected objects were subsequently manually corrected, and the total number of objects per nucleus was recorded.

Tracking and quantification of live movies was performed using the Trackmate plugin of Fiji/ImageJ and the Track Manager of Icy. Movies were filtered using a mean filter size of 1 pixel, and maximum intensity projections were generated using Fiji/ImageJ. Drift in x/y was corrected using the registration function correct 3D drift of Fiji/ImageJ and IN.SNAP/OR3 objects were detected using a LoG (Laplacian of Gaussian) filter and tracked using the simple LAP tracker or manually. Tracks were exported to Icy and with the Track Manager plugin the local background corrected mean intensities were calculated using the intensity profiler track processor with a disk diameter of 2 pixels (440 nm) and a local background ring diameter of 2–5 pixels (440–1100 nm). Distances between tracks were calculated using the distance profiler track processor. Intensities were normalized to the highest value and plotted using the statistical visualization toolbox seaborn v0.13.0 (Waskom, 2021) in Python.

### CLEM sample preparation and electron tomography

$1.2 \times 10^5$ TZM-bl mScarlet.OR3 cells or $4 \times 10^4$ MDM were seeded on carbon-coated 3 mm sapphire discs in a 35 mm glass-bottom dish (MatTek, USA), on the next day infected with VSV-G pseudotyped IN.SNAP.SiR-labeled NNHIV-ANCH (30 μU RT/cell for TZM-bl cells and 60 μU RT/cell for MDM), and incubated for 24 (TZM-bl) or 72 (MDM) hr at 37 °C. TZM-bl cells were treated with 500 nM LEN, 15 μM PF74, or DMSO vehicle for 1 h prior to cryo-immobilization by high-pressure freezing. Subsequent sample processing and electron tomography were performed as described previously (Müller et al, 2021; Kreysing et al, 2025).

### Integration assay by Alu-PCR

Differentiated MDM ($1.2 \times 10^6$ monocytes per sample seeded in 10-cm dishes with 6 ml medium for 1 week) were infected with VSV-G pseudotyped HIV-1 NL4-3 (15 μUnits RT/cell) and incubated at 37 °C

and 5% $CO_2$. 70 h after infection, cells were treated with 500 nM LEN or equal volumes of DMSO in the presence of 20 μM Efavirenz (EFV). Cells were harvested for total genomic DNA extraction 1 or 4 h after treatment using the Zymo Research Quick-DNA Miniprep Kit (Zymo Research), following the manufacturer's instructions. The relative amount of integrated HIV-1 viral DNA (vDNA) was quantified by nested Alu-PCR (Tan et al, 2006). Briefly, integrated vDNA was first amplified from 100 ng of genomic DNA, using a primer targeting a conserved region of Alu elements (Alu1) and a second primer annealing to HIV-1 LTRs (LM667) containing a phage lambda-specific (λT) sequence on its 5' end. In the second nested quantitative PCR (qPCR), products of the first PCR were diluted (1:500) and amplified using iQ™ SuperMix (BioRad), λT-specific primer, LR internal LTR primer, and a TaqMan probe ZXF-P (Reagents and Tools Table). In a parallel reaction, 10 ng of genomic DNA was used to amplify gene lamin B2 (LMNB2) B13 region with specific primers and a TaqMan probe (Reagents and Tools Table) which served to normalize relative integration levels to the cell number by the ΔΔCt method (Livak and Schmittgen, 2001). qPCR reactions were performed on a CFX96 C1000 Touch Thermal Cycler (BioRad) with the following conditions: 98 °C 3 min; 45 cycles of 98 °C 10 s, 60 °C 40 s; 98 °C 10 min.

## Data availability

This study includes no data deposited in external repositories.

The source data of this paper are collected in the following database record: biostudies:S-SCDT-10_1038-S44318-025-00652-5.

## Peer review information

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

## Acknowledgements

We are grateful to M. Kvaratskhelia, University of Colorado for a kind gift of LEN. The ANCHOR system is developed by and available from NeoVirTech (France, http://www.neovirtech.com). We acknowledge the microscopy support from the Infectious Diseases Imaging Platform (IDIP) of the Center for Integrative Infectious Disease Research, Heidelberg. For the publication fee we acknowledge financial support by Heidelberg University. This work was funded by the Deutsche Forschungsgemeinschaft (DFG, German Research Foundation)—Projektnummer 240245660—SFB 1129 project 5 (H-GK), project 6 (BM), project 4 (USS), project 20 (ML), and by the TTU HIV in the DZIF (VL, H-GK).

## Author contributions

**Thorsten G Müller**: Conceptualization; Data curation; Formal analysis; Validation; Investigation; Visualization; Methodology; Writing—original draft; Writing—review and editing. **Severina Klaus**: Data curation; Formal analysis; Investigation; Visualization; Methodology; Writing—original draft; Writing—review and editing. **Vojtech Zila**: Formal analysis; Validation; Investigation; Visualization; Methodology; Writing—original draft; Writing—review and editing. **Bojana Lucic**: Investigation; Methodology; Writing—review and editing. **Carlotta Penzo**: Investigation; Methodology; Writing—review and editing. **Svenja L Nopper**: Investigation; Writing—review and editing. **Gonen Golani**: Investigation; Writing—review and editing. **Maria Anders-Össwein**: Investigation. **Vera Sonntag-Buck**: Investigation. **Anke-Mareil Heuser**: Investigation. **Ulrich S Schwarz**: Supervision; Funding acquisition; Validation; Project administration; Writing—review and editing. **Vibor Laketa**: Investigation; Writing—review and editing. **Marina Lusic**: Supervision; Funding acquisition; Methodology; Project administration; Writing—review and editing. **Barbara Müller**: Supervision; Funding acquisition; Validation; Writing—original draft; Project administration; Writing—review and editing. **Hans-Georg Kräusslich**: Conceptualization; Supervision; Funding acquisition; Validation; Writing—original draft; Project administration; Writing—review and editing.

Source data underlying figure panels in this paper may have individual authorship assigned. Where available, figure panel/source data authorship is listed in the following database record: biostudies:S-SCDT-10_1038-S44318-025-00652-5.

## Funding

## Disclosure and competing interests statement

The authors declare no competing interests.

# Expanded View Figures

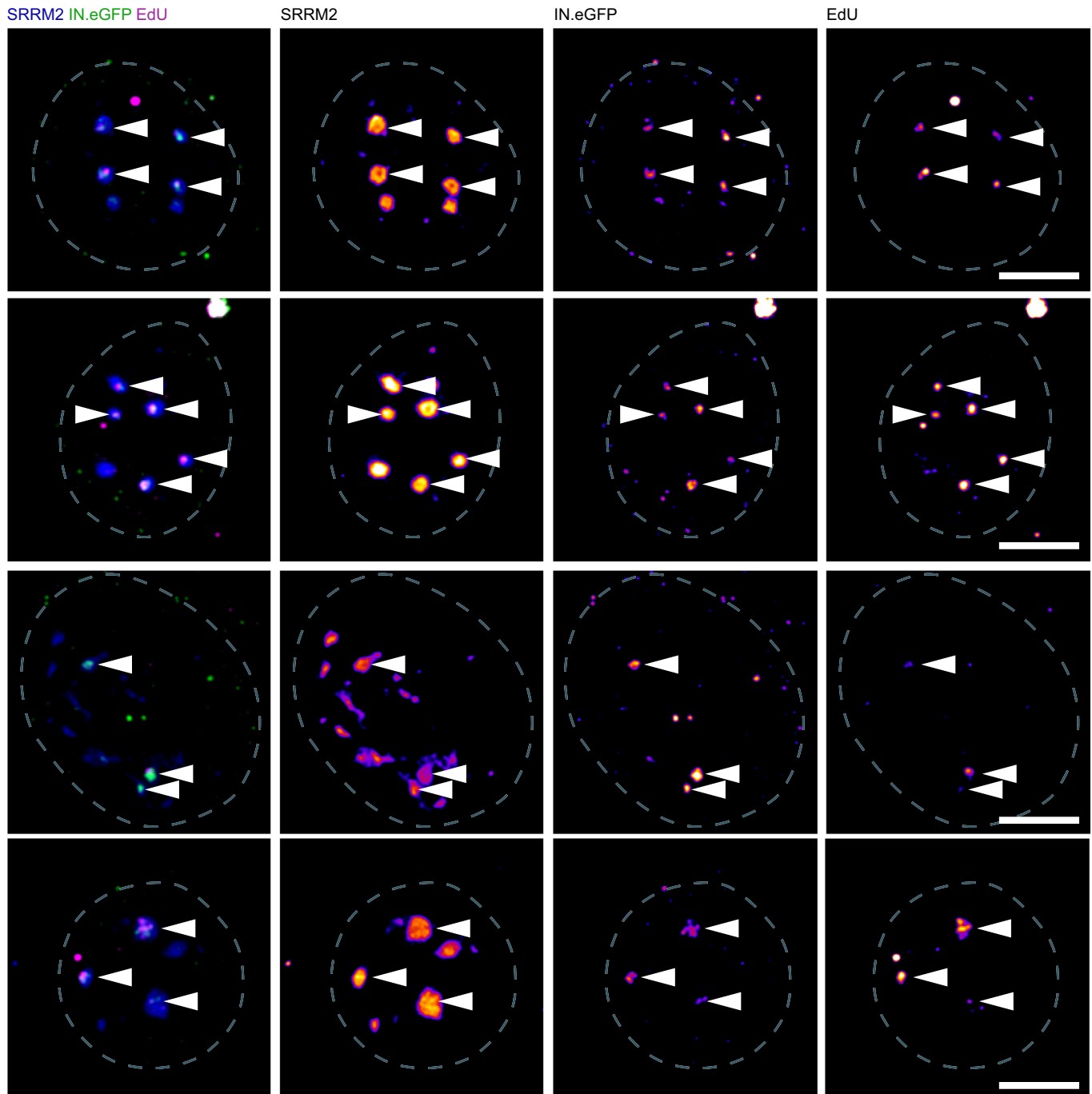

**Figure EV1.  Additional examples of subviral complexes in nuclear speckles of primary MDM.**

Super-resolution analysis of HIV-1 cDNA within nuclear speckles of monocyte-derived macrophages (MDM) showing EdU (magenta) and IN.eGFP signals (green) in the center of SRRM2 condensates (blue). Shown are four additional maximum intensity projection of MDM nuclei (white dashed line) infected for 72 h with VSV-G pseudotyped NNHIV in presence of EdU followed by fixation, EdU click labeling and immunofluorescence staining using an antibody against SRRM2 (SC35). White arrowheads indicate a selection of nuclear IN.eGFP objects for clarity. Samples were imaged using Airyscan microscopy. Scale bars: 5 µm.

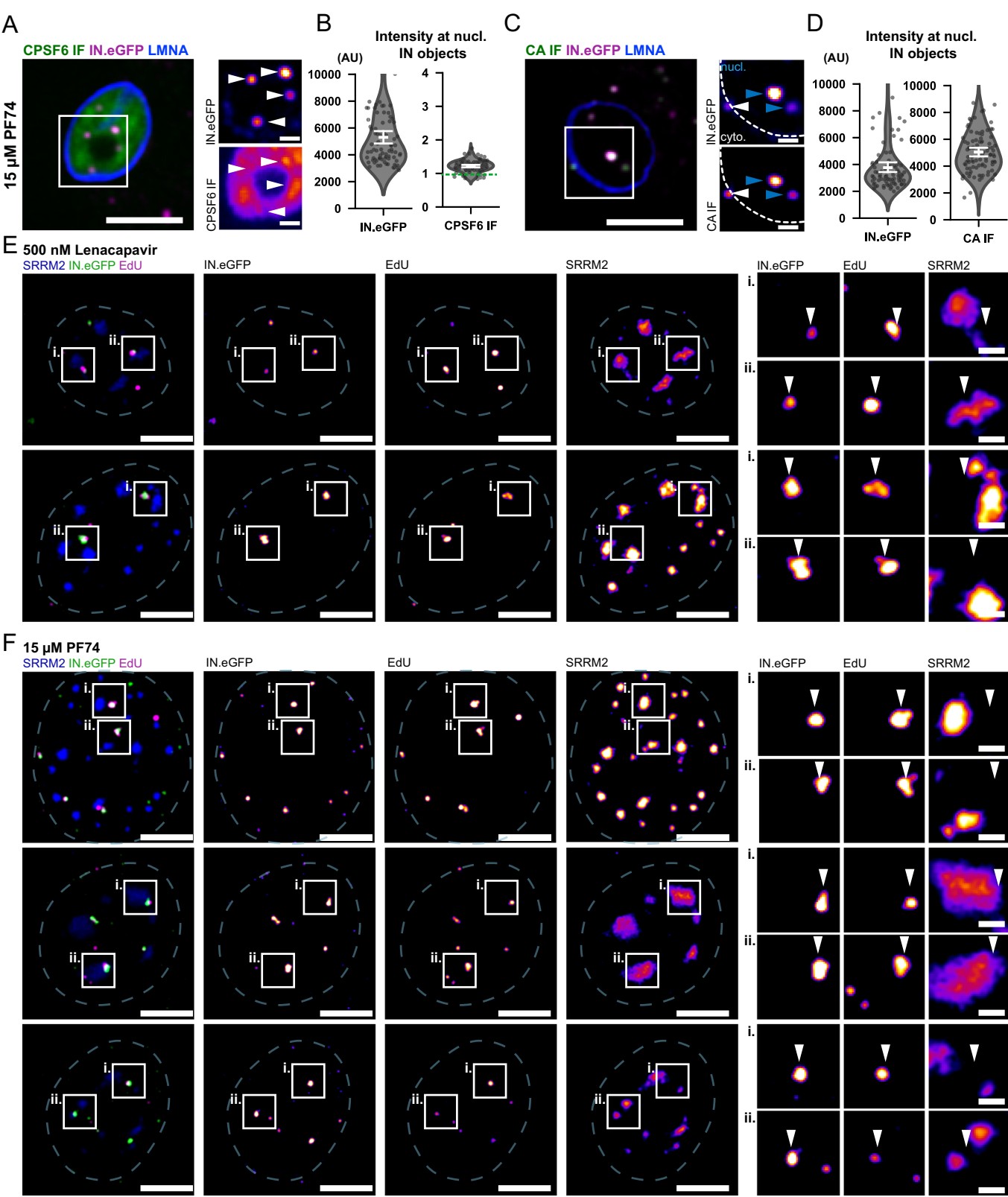

**Figure EV2. Effect of PF74 on CA and CPSF6 signals in MDM and additional examples of LEN and PF74 induced subviral particle exit from nuclear speckles.**

(A–F) MDM were infected using IN.eGFP labeled VSV-G pseudotyped NNHIV IN.SNAP for 72 h before addition of indicated concentrations of PF74 or LEN for 1 h. Cells were fixed and immunostained before 3D SDCM imaging (**A–D**) or 3D Airyscan imaging (**E, F**). Samples were stained for CPSF6 (**A, B**), CA (**C, D**) or SRRM2 (**E, F**). Maximum intensity projections are shown. Error bars represent SEM. Scale bars: 5 µm (overviews) and 1 µm (enlargements). (**A–D**) Stripping of pre-assembled CPSF6 (**A, B**) and exposure of masked CA epitopes (**C, D**) by 15 µm PF74. (**A, E, F**) White arrowheads indicate a selection of nuclear IN.eGFP objects for clarity. (**C**) White arrowheads indicate cytoplasmic IN.eGFP objects whereas blue arrowheads indicate nuclear IN.eGFP objects. Dotted lines indicate nuclear boundary. (**B, D**) Images were analyzed by automated quantification using custom-made Python code as described in the Methods and protocols section. CPSF6 signals (**B**) were normalized to the mean nuclear CPSF6 expression level of the respective cell (green dotted line at $y = 1$). (**E, F**) Displacement of IN.eGFP objects from nuclear speckles. Cells were infected in presence of 10 µm EdU and click labeled prior to immunofluorescence staining. Shown are two representative cell nuclei treated with 500 nM LEN for 1 h (**E**) and three nuclei treated with 15 µM PF74 for 1 h (**F**).

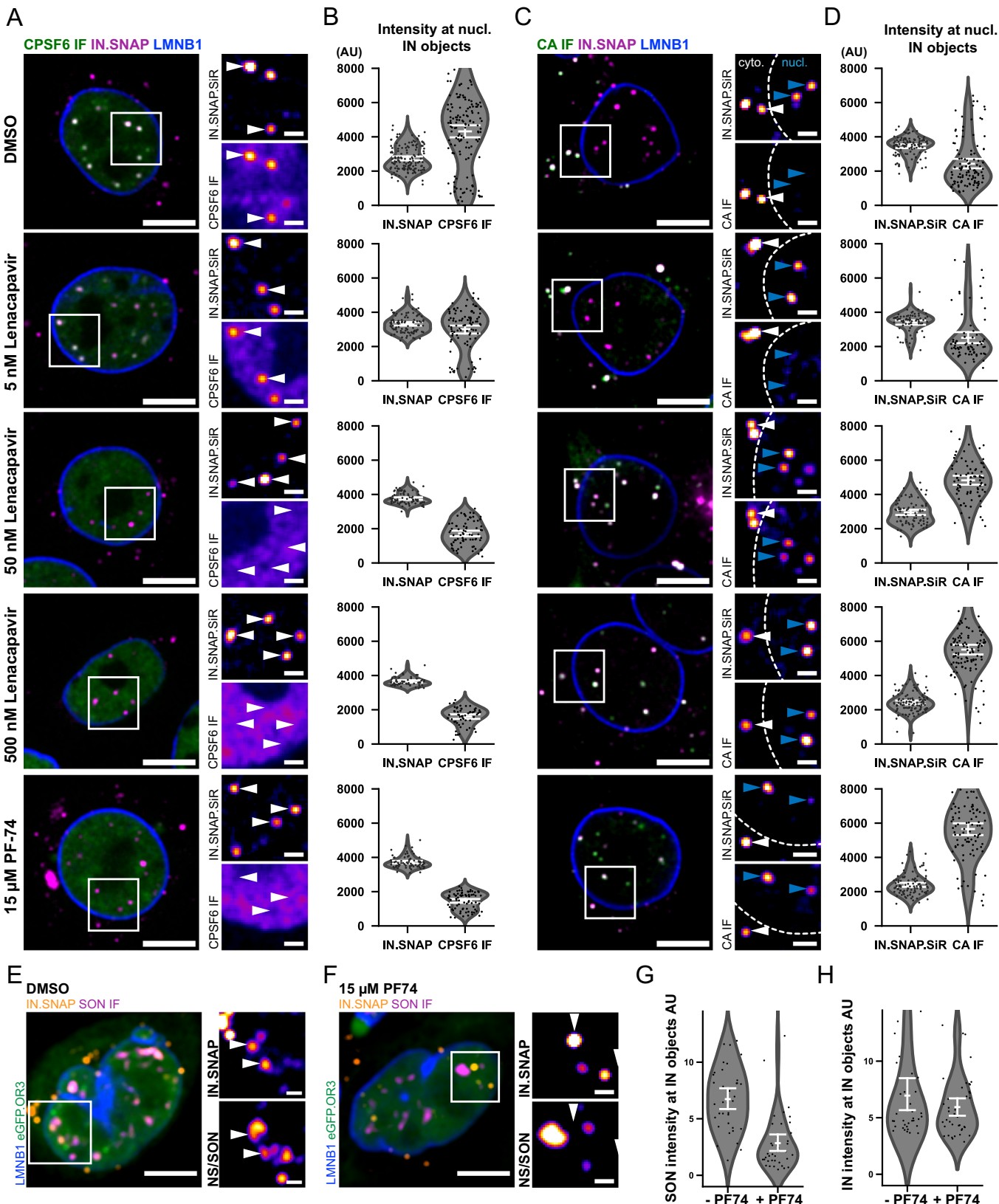

◀

**Figure EV3.  Lenacapavir and PF74 strip CPSF6 from nuclear capsids, concomitantly expose masked CA epitopes, and lead to capsid exit from nuclear speckles in HeLa-based TZM-bl cells.**

(A–H) Cells were infected using IN.SNAP labeled VSV-G pseudotyped NNHIV for 24 h before addition of indicated amounts of Lenacapavir, PF74 or DMSO for 1 h. Cells were fixed and immunostained before 3D confocal spinning disc imaging. Samples were stained for CPSF6 (**A**, **B**), CA (**C**, **D**) or SON (**E–H**). Shown is one of three independent experiments. Error bars represent SEM. Scale bars: 5 μm (overviews) and 1 μm (enlargements). (**A–D**) Stripping of pre-assembled CPSF6 (**A**, **B**) and concomitant exposure of masked CA epitopes (**C**, **D**) by Lenacapavir and PF74. (**A**, **E**, **F**) White arrowheads indicate a selection of nuclear IN.SNAP objects for clarity. (**C**) White arrowheads indicate cytoplasmic IN.SNAP objects whereas blue arrowheads indicate nuclear IN.SNAP objects. (**B**, **D**) Images were analyzed by automated quantification using custom-made Python code. Nuclear 3D IN.SNAP objects were segmented and mean intensities in the respective channels quantified. (**E–H**) Displacement of IN.SNAP objects from nuclear speckles. Shown are representative cells treated with DMSO (**E**) or 15 μM PF74 (**F**) for 1 h. (**G**, **H**) Quantification of SON (**G**) or IN.SNAP (**H**) mean intensities shown in (**E**, **F**) at nuclear 3D IN.SNAP objects in presence and absence of 15 μM PF74.

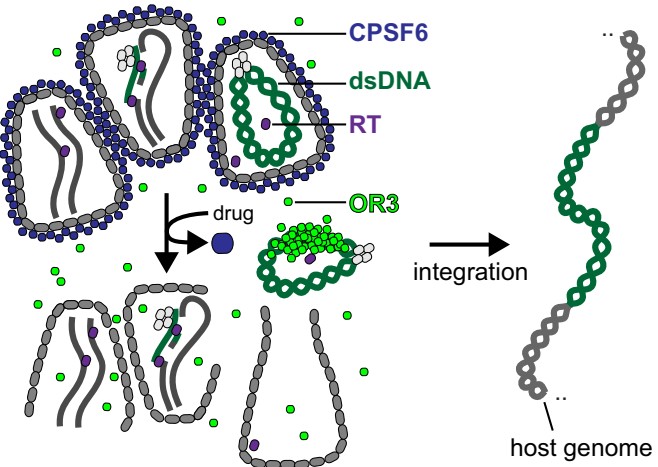

**Figure EV4.  Scheme of LEN/PF74 induced exposure of HIV-1 dsDNA.**

LEN or PF74 displace CPSF6 from HIV-1 capsids clustered within nuclear speckles, thereby relocalizing these complexes to speckle-adjacent sites. Concomitantly, capsid structures are damaged, with bifurcated protrusions appearing at the narrow end of capsids. Functionally complete reverse-transcribed HIV-1 genomes are released and integrate into the host-cell chromatin.

