## [Peer Review File · The EMBO Journal]

Lenacapavir-induced capsid damage uncovers HIV-1 genomes emanating from nuclear speckles

Thorsten Müller, Severina Klaus, Vojtech Zila, Bojana Lucic, Carlotta Penzo, Svenja Nopper, Gonen Golani, Maria Anders-Össwein, Vera Sonntag-Buck, Anke-Mareil Heuser, Ulrich Schwarz, Vibor Laketa, Marina Lusic, Barbara Müller, and Hans-Georg Kräusslich

Corresponding author(s): Thorsten Müller (muellert@ie-freiburg.mpg.de), Hans-Georg Kräusslich (Hans-Georg.Kraeusslich@med.uni-heidelberg.de)

Review Timeline:

Submission Date:	8th Jul 25
Editorial Decision:	2nd Sep 25
Revision Received:	17th Oct 25
Editorial Decision:	7th Nov 25
Revision Received:	10th Nov 25
Accepted:	13th Nov 25

Editor: Hartmut Vodermaier

Transaction Report:

Dr Hans-Georg Krausslich
Heidelberg, Ruprecht-Karls-Universitt
Abteilung Virologie
Universitt Heidelberg
Im Neuenheimer Feld 324
Heidelberg D69120
Germany

2nd Sep 2025

Re: EMBOJ-2025-121832

Lenacapavir-induced capsid damage exposes HIV-1 genomes from nuclear speckles.

Dear Dr. Krausslich,

Thank you for submitting your manuscript on Lenacapavir effects on nuclear HIV-1 core uncoating. I apologize for the considerable delay in getting back to you with a decision - we sent your study to three expert referees, but have so far (despite multiple reminders sent from our office) still only received the reports of two of them. Since both of them are in fair agreement, and to prevent further loss of time, I have chosen to contact you now with an invitation to start revising the study based on the overall supportive comments copied below. As you will see, referee 2's points are mostly focused on presentational aspects, while referee 3 additionally raises important concerns about insufficient functional follow-up investigation. Should you be able to adequately improve this aspect, for example by heeding the well-taken experimental suggestions of the reviewer, we would be pleased to consider a revised manuscript further for publication.

Please note that this remains at present still a preliminary decision, and that any specific issues raised by the outstanding third report (should it arrive within the next two weeks) may still need to be incorporated before finalization and resubmission of the work. I should also point out that our policy to allow only a single round of (major) revision makes it important to carefully respond to all points raised at this point. As always, competing work appearing during the course of the revision period will not affect our final decision on your study. Should you have any further questions linked to this decision, the referee report or the revision guidelines, please do not hesitate to contact me.

Thank you again for the opportunity to consider this work for The EMBO Journal, and I shall be in contact shortly to either send you the outstanding third report, or to finalize the decision without it.

Yours sincerely,

Hartmut Vodermaier

9) To facilitate reproducibility and cross-laboratory adoption of methodologies, please structure the Materials & Methods section as outlined in our guide to authors, including a completed Reagents and Tools Table that can be downloaded from our author guidelines as well (<https://www.embopress.org/page/journal/14602075/authorguide#structuredmethods>).

10) Digital image enhancement is acceptable practice, as long as it accurately represents the original data and conforms to community standards. If a figure has been subjected to significant electronic manipulation, this must be clearly noted in the figure legend and/or the 'Materials and Methods' section. The editors reserve the right to request original versions of figures and the original images that were used to assemble the figure. Finally, we generally encourage uploading of numerical as well as gel/blot image source data; for details see: embopress.org/page/journal/14602075/authorguide#sourcedata

In the interest of ensuring the conceptual advance provided by the work, we recommend submitting a revision within 3 months (1st Dec 2025). Please discuss the revision progress ahead of this time with the editor if you require more time to complete the revisions. Use the link below to submit your revision:

Link Not Available

Referee #2:

This study uses cutting edge approaches including CLEM, ET, STED and ANCHOR to characterize how high concentrations of HIV-1 capsid inhibitors lenacapavir (LEN) and PF74 affect uncoating of viral cores in the nucleus. These findings are not surprising, but they are of sufficient general interest to be published in a high-impact journal. I recommend the following improvements.

1. The abstract states: "...speckle-associated factors appear to be required to promote genome uncoating." I strongly suggest rephrasing this sentence because there is no evidence presented in the present study to indicate that the speckle proteins "promote" uncoating. In fact, I like their discussion, where they state: "Drug induced displacement of the CPSF6 coat could make the capsid more sensible to breakage by dsDNA or could expose a previously broken capsid lattice that had been stabilized by the outer CPSF6 coat." In other words, CPSF6 could potentially be counteracting or shielding the uncoating process, whereas reverse transcribed dsDNA could be a primary force behind breaking HIV-1 cores. Accordingly, I suggest stating in the abstract that speckle-associated factors could regulate uncoating of HIV-1 cores in the nucleus.

2. The title of the sections of Results is: "Nuclear speckle localization of capsids can be pharmacologically perturbed using LEN". Remove "pharmacologically", which implies that this is a primary antiviral mechanism of action of LEN. In this section they nicely

cite the previous work to state that LEN EC50 values are ~50 pM in several cell lines. Yet, ~10,000-fold higher concentration (500 nM) of LEN was needed to displace CPSF6 from nuclear subviral complexes in MDMs (Figure 2). Similarly, for the CLEM-ET experiments in Figure 5 they used 500 nM LEN. There is nothing wrong with using such high concentrations of LEN as an investigational probe to study HIV-1 biology but avoid using terms such as "pharmacological". Furthermore, I suggest clarifying once again in Discussion that observed capsid alterations were detected at LEN concentrations that substantially exceed antiviral EC50 values of the inhibitor.

3. A previous study (PMID: 36202818) examined the CPSF6 displacement from HIV-1 cores by LEN in TZM-bl cells and the reasons for the need of high LEN concentrations have been discussed in the indicated report. The present studies corroborate with the previous findings. However, PMID: 36202818, which is highly relevant to the present study, is not referenced or discussed.

Referee #3:

The manuscript by Mueller et al. investigates the pre-integration steps in the HIV-1 life cycle and specifically ask the question if reverse transcription of the viral genome, which is completed in the nucleus, is the key driver of uncoating of the viral core. The authors use elegant imaging techniques, combining super-resolution microscopy with correlative light and electron microscopy, and label different components of the virus to study a narrow window of time during the late steps of reverse transcription in the nucleus of primary macrophages. In agreement with previous reports, the authors localized HIV-1 capsid core clusters in nuclear speckles and showed that treatment with drugs lenacapavir (LEN) or PF74, which bind and alter the viral core structure, displaced host-cofactor CPSF6 from the capsid core. In addition, these drugs removed the viral cores from speckles simultaneously breaking them. Such damaged cores appeared to release nucleic acids from the narrow end, becoming more permeable to fluorescent molecules that label the viral genome once reverse transcription is nearly completed. The authors conclude that the late steps of reverse transcription take place in an intact core coated by CPSF6 in nuclear speckles and that LEN perturbs these normal events by prematurely breaking the cores.

The manuscript addresses an interesting topic, which is poorly understood. There is some controversy on the role of reverse transcription in uncoating, with some studies suggesting that the conversion of RNA into bulkier DNA is sufficient for uncoating and others suggesting that there are host-cofactors that help. The results shown here are of good quality and well-presented and the manuscript is clearly written. However, although some of the results are novel, the manuscript does not provide much information on the putative uncoating factor and lacks functional follow up.

1. One question that remains unaddressed is if the breaking of the cores in the nucleus by LEN affects function. The authors have a nice experimental set-up that is timed to detect the late steps of reverse transcription in the nucleus. They know that in these conditions nuclear import and reverse transcription are unaffected by the drugs. But is integration impaired? Can the breaking of cores in the nucleus be correlated with efficiency of integration? Perhaps this can be tested by Alu-LTR qPCR, or perhaps even by imaging.
2. A recent study (Ay et al. EMBO J. 2025 PMID: 39623137), which is cited by the authors, also shows that CPSF6 condensates protect and stabilize viral cores but Ay et al. found that reverse transcription reduces the number of apparently intact cores and, notably, observe similar filamentous-like structures protruding from their narrow end. Their interpretation is that reverse transcription per se induces uncoating. This discrepancy should be discussed.
3. Can the results (more cores with positive labelling for late DNA transcripts) also be interpreted to suggest that breaking of the cores promotes more efficient reverse transcription?
4. Figure 2e. It would be helpful to show the quantification of these images showing displacement of cores from speckles.

Referee #2:

This study uses cutting edge approaches including CLEM, ET, STED and ANCHOR to characterize how high concentrations of HIV-1 capsid inhibitors lenacapavir (LEN) and PF74 affect uncoating of viral cores in the nucleus. These findings are not surprising, but they are of sufficient general interest to be published in a high-impact journal. I recommend the following improvements.

1. The abstract states: "...speckle-associated factors appear to be required to promote genome uncoating." I strongly suggest rephrasing this sentence because there is no evidence presented in the present study to indicate that the speckle proteins "promote" uncoating. In fact, I like their discussion, where they state: "Drug induced displacement of the CPSF6 coat could make the capsid more sensible to breakage by dsDNA or could expose a previously broken capsid lattice that had been stabilized by the outer CPSF6 coat." In other words, CPSF6 could potentially be counteracting or shielding the uncoating process, whereas reverse transcribed dsDNA could be a primary force behind breaking HIV-1 cores. Accordingly, I suggest stating in the abstract that speckle-associated factors could regulate uncoating of HIV-1 cores in the nucleus.

We changed the wording as suggested to "*speckle-associated factors could regulate genome uncoating*".

2. The title of the sections of Results is: "Nuclear speckle localization of capsids can be pharmacologically perturbed using LEN". Remove "pharmacologically", which implies that this is a primary antiviral mechanism of action of LEN. In this section they nicely cite the previous work to state that LEN EC50 values are ~50 pM in several cell lines. Yet, ~10,000-fold higher concentration (500 nM) of LEN was needed to displace CPSF6 from nuclear subviral complexes in MDMs (Figure 2). Similarly, for the CLEM-ET experiments in Figure 5 they used 500 nM LEN. There is nothing wrong with using such high concentrations of LEN as an investigational probe to study HIV-1 biology but avoid using terms such as "pharmacological". Furthermore, I suggest clarifying once again in Discussion that observed capsid alterations were detected at LEN concentrations that substantially exceed antiviral EC50 values of the inhibitor.

We agree with the reviewer and have removed the word "pharmacologically" from the section heading. In the Discussion we added the following sentence to clarify the effect of different Lenacapavir concentrations: "*These alterations were detected at LEN concentrations that remove the CPSF6-layer from the hydrophobic binding pocket, and which substantially exceed antiviral EC50 values of the inhibitor.*"

3. A previous study (PMID: 36202818) examined the CPSF6 displacement from HIV-1 cores by LEN in TZM-bl cells and the reasons for the need of high LEN concentrations have been discussed in the indicated report. The present studies corroborate with the previous findings. However, PMID: 36202818, which is highly relevant to the present study, is not referenced or discussed.

We thank the reviewer for pointing out this relevant oversight. In the revised manuscript, we added the citation to the following sentence: "*We hypothesize that not only the immersion of capsids into the phase-separated speckle environment (Xu et al, 2022), but also this specific*

subdomain localization is driven by the biophysical properties of the multimeric CPSF6 coat (Wei et al, 2022) encasing nuclear capsids.”

We now also discuss this report in the Discussion section as follows: *“Anti-viral LEN activity at lower concentrations was previously attributed to binding of LEN to unoccupied hydrophobic pockets without displacing cellular cofactors (Wei et al, 2022).”*

Referee #3:

The manuscript by Mueller et al. investigates the pre-integration steps in the HIV-1 life cycle and specifically ask the question if reverse transcription of the viral genome, which is completed in the nucleus, is the key driver of uncoating of the viral core. The authors use elegant imaging techniques, combining super-resolution microscopy with correlative light and electron microscopy, and label different components of the virus to study a narrow window of time during the late steps of reverse transcription in the nucleus of primary macrophages. In agreement with previous reports, the authors localized HIV-1 capsid core clusters in nuclear speckles and showed that treatment with drugs lenacapavir (LEN) or PF74, which bind and alter the viral core structure, displaced host-cofactor CPSF6 from the capsid core. In addition, these drugs removed the viral cores from speckles simultaneously breaking them. Such damaged cores appeared to release nucleic acids from the narrow end, becoming more permeable to fluorescent molecules that label the viral genome once reverse transcription is nearly completed. The authors conclude that the late steps of reverse transcription take place in an intact core coated by CPSF6 in nuclear speckles and that LEN perturbs these normal events by prematurely breaking the cores.

The manuscript addresses an interesting topic, which is poorly understood. There is some controversy on the role of reverse transcription in uncoating, with some studies suggesting that the conversion of RNA into bulkier DNA is sufficient for uncoating and others suggesting that there are host-cofactors that help. The results shown here are of good quality and well-presented and the manuscript is clearly written. However, although some of the results are novel, the manuscript does not provide much information on the putative uncoating factor and lacks functional follow up.

1. One question that remains unaddressed is if the breaking of the cores in the nucleus by LEN affects function. The authors have a nice experimental set-up that is timed to detect the late steps of reverse transcription in the nucleus. They know that in these conditions nuclear import and reverse transcription are unaffected by the drugs. But is integration impaired? Can the breaking of cores in the nucleus be correlated with efficiency of integration? Perhaps this can be tested by Alu-LTR qPCR, or perhaps even by imaging.

We thank the reviewer for suggesting this important experiment. Following the reviewer’s suggestion, we have performed HIV-1 infection experiments in primary MDM from three independent donors and quantified the amount of integrated HIV-1 genomes following LEN treatment using Alu-LTR qPCR. MDM infected with wild-type HIV-1 were treated with LEN or DMSO for 1h or 4h at 70h post infection, similar to the previously shown experiments where the integration-deficient variant NNHIV had been used. To avoid further reverse transcription during the period of drug addition, the RT inhibitor Efavirenz was added to all samples together with LEN or DMSO.

This experiment showed a twofold increase of integrated genomes both after 1h and 4h incubation with LEN compared to the control. These results are presented in the revised manuscript as new panel G in Fig. 4, and we have pasted this panel below.

We have also added the respective methodology to the Methods section. Furthermore, to perform these challenging experiments within the short revision period, we turned to our colleagues in the institute with long experience in Alu-LTR qPCR. Because of their important contributions, we have expanded the list of authors accordingly.

Importantly, the results of this additional experiment confirmed that functionally complete and integration-competent HIV-1 cDNA genomes can be released from capsids not accessible to the fusion protein prior to LEN treatment. This conclusion could not be drawn from the ANCHOR detection using integration-defective viruses (as used throughout the manuscript) since the recognition sequence does not require full-length cDNA synthesis. Accordingly, we had previously written that complete or almost complete genomes could be released by LEN from closed capsid structures. This has now been revised based on the new data, and we are very grateful to the reviewer for leading us into this direction.

The results of this experiment are now included in the Results section of the manuscript and put into context in the Discussion:

Results: “To expand this analysis to MDM and test whether HIV-1 cDNA revealed by LEN from CPSF6-coated nuclear capsids is competent for integration into host cell chromatin, we performed quantitation of integrated HIV-1 proviral sequences in LEN- or control-treated MDM using Alu-LTR qPCR. Since NNHIV is not competent for integration, we infected primary MDM from three different donors with VSV-G pseudotyped NL4-3 wild-type virus for 70 h before treating the cells with LEN or DMSO for 1h or 4h, respectively. To exclude the possibility that a change in the number of integration events after drug treatment could be due to continued nuclear import and reverse transcription, thereby confounding the result, we added EFV to all samples at the time of DMSO or LEN treatment, effectively blocking further reverse transcription. LEN treatment for 1h or 4h of LEN treatment led to a ca. twofold increase of Alu-LTR qPCR products (Fig. 4G), indicating that integration-competent, functionally complete proviral HIV-1 cDNA could be revealed from CPSF6-coated nuclear capsids by this treatment.”

Discussion: “Furthermore, LEN treatment also induced a twofold increase of integrated HIV-1 genomes when primary MDM were infected by wild-type virus. This effect did not require ongoing reverse transcription indicating that functionally complete and integration-competent genomic

HIV-1 DNA had been synthesized inside nuclear capsids and could be revealed by LEN treatment. These observations are consistent with the ability of HIV-1 to evade activation of innate immune responses (Cingöz & Goff, 2019) unless cGAS-mediated sensing is induced by PF74 (Sumner et al, 2020; Ay et al, 2025) or LEN (Scott et al, 2025).

Our findings are consistent with reports suggesting that synthesis of long dsDNA inside the HIV-1 capsid is important for capsid breakage and genome uncoating, but suggest that reverse transcription may not be sufficient. ANCHOR detection in our system requires the dsDNA to be completed to at least 7.3 kbp. This DNA length is well above the critical limit of ~ 6 kbp dsDNA for efficient HIV-1 genome uncoating reported in a previous study (Burdick et al, 2024). Furthermore, the observed increase of integration events as shown by Alu-LTR qPCR and independent of ongoing reverse transcription shows that at least some of the nuclear capsids contain functionally complete viral genomes that were not accessible prior to LEN treatment.”

2. A recent study (Ay et al. EMBO J. 2025 PMID: 39623137), which is cited by the authors, also shows that CPSF6 condensates protect and stabilize viral cores but Ay et al. found that reverse transcription reduces the number of apparently intact cores and, notably, observe similar filamentous-like structures protruding from their narrow end. Their interpretation is that reverse transcription per se induces uncoating. This discrepancy should be discussed.

Ay et al. used Immuno-Gold EM analysis of the infected THP-1 cell line in the presence or absence of the RT inhibitor NEV. They did not, however, analyse the effect of Lenacapavir or PF74 on the viral ultrastructure. Note that Immuno Gold-EM uses an extraction procedure that may impede the structural integrity of the sample. The authors showed an example of an empty core (they termed it “ghost”) with electron density in close vicinity to the core. This diffuse density seems to be at the wide end of the core (see panel C below). In comparison, we observed, upon treatment with LEN, clear protrusions from the narrow ends of cones. We discussed what these could represent in the discussion section of the manuscript.

LEN treatment in our experiments consistently revealed elongated protrusions at the narrow end (Figure 5C right panel). Importantly, our observations are not in conflict with previous studies reporting that reverse transcription of sufficiently long cDNA is needed for efficient uncoating. We specifically state in our manuscript that the process of reverse transcription appears to be necessary, but - based on the data shown in the current report - may not be sufficient to induce immediate uncoating. In the Discussion section we point out the similarities between our work and the previous report by Ay et al. *“The observation of broken capsids without interior density is consistent with prior results in different cell lines that also reported incomplete capsid lattice remnants inside the nucleus of HIV-1 infected cells (Müller et al, 2021; Ay et al, 2025).”* For the discussion of reverse transcription of a sufficiently long dsDNA molecule being necessary for uncoating, we are referencing the report by Burdick et al., which provided direct and functional evidence analyzing HIV-1-based vectors of different genome lengths.

3. Can the results (more cores with positive labelling for late DNA transcripts) also be interpreted to suggest that breaking of the cores promotes more efficient reverse transcription?

We tested this hypothesis experimentally in Figure 4. The increased number of dsDNA/ANCHOR signals upon LEN treatment was also observed when further reverse transcription during treatment with capsid-targeting drugs was blocked by addition of an RT inhibitor, indicating that this increase was not due to promotion of reverse transcription. We have tried to make this clearer in the revised version by more explicitly stating that this hypothesis was experimentally tested and added the following sentence to the Results section.

“The appearance of new OR3 signals colocalizing with HIV-1 subviral complexes could also be explained if capsid breakage would promote more efficient reverse transcription, or if previously stalled reverse transcription complexes would resume reverse transcription to complete dsDNA upon opening of the capsid.” This is then followed by the experimental result.

4. Figure 2e. It would be helpful to show the quantification of these images showing displacement of cores from speckles.

Following this suggestion by the reviewer, we have performed additional analyses and present the quantification in panel F of Fig. 2 in the revised version of the manuscript.

Dr. Thorsten G. Müller
Heidelberg University
Department of Infectious Diseases, Virology
Heidelberg 69120
Germany

7th Nov 2025

Re: EMBOJ-2025-121832R
Lenacapavir-induced capsid damage exposes HIV-1 genomes from nuclear speckles.

Dear Drs. Müller and Kräusslich,

Thank you for submitting your revised manuscript to The EMBO Journal. It has now been re-reviewed by the two original referees, who were both fully satisfied with the revisions. We shall therefore be happy to proceed with acceptance and publication of the study, as soon as the following editorial issues have also been adequately dealt with:

- Please consider a more accessible title for our broad general readership. In particular, I am not sure that one can "expose (something) from..." in English. A clearer phrasing may be e.g.

"Lenacapavir-induced capsid damage uncovers HIV-1 genomes emanating from nuclear speckles"?
Happy to discuss!

- On the abstract page, please pre-face the keywords with the header "Keywords", and please rename the Material and Methods into simply "Methods"

- Please carefully go through the reference list and make sure that each reference is complete with citation year, volume, and page/locator numbers - such information is currently missing for some of them. Also, please adjust the format for citation of preprints as specified in our author guidelines:

The citation in the text should be: "(preprint: NAME1 et al, YEAR)"

The citation in the reference list: "NAME1, NAME2, ... (YEAR) [article]. bioRxiv doi: XXX"

- Please upload each EV movie file packed together with its respective EV Movie title/legend text file into individual ZIP archives, one per movie.

- Finally, during routine pre-acceptance checks, our data editors have raised the following queries regarding figures, data, and legends; I would appreciate if you briefly answered to them in the cover letter of your final submission, and made the requested text modifications with changes/additions highlighted via the "Track changes" option, to facilitate our final checking.

1. Please note that the measure of center for the error bars needs to be defined in the legends of figures 4D-G
2. Please note that the dashed lines are not defined in the legend of figure 1B. This needs to be rectified.
3. Please note that the white arrow heads are not defined in the legend of figure EV1, EV2 A, E, F; EV3 A, C, E, F. This needs to be rectified.
4. Please note that the circles are not defined in the legend of figure 3E, G. This needs to be rectified.

I am returning the manuscript to you for a final round of minor revision, solely to allow you to make these modifications and upload the revised files. Once we will have received them, we should be ready to swiftly proceed with formal acceptance and production of the manuscript.

Yours sincerely,

Hartmut Vodermaier

- 1) Every manuscript requires a Data Availability section (even if only stating that no deposited datasets are included). Primary datasets or computer code produced in the current study have to be deposited in appropriate public repositories prior to resubmission, and reviewer access details provided in case that public access is not yet allowed. Further information: embopress.org/page/journal/14602075/authorguide#dataavailability
- 2) Each figure legend must specify
 - size of the scale bars that are mandatory for all micrograph panels
 - the statistical test used to generate error bars and P-values
 - the type error bars (e.g., S.E.M., S.D.)
 - the number (n) and nature (biological or technical replicate) of independent experiments underlying each data point
 - Figures may not include error bars for experiments with $n < 3$; scatter plots showing individual data points should be used instead.
- 3) Revised manuscript text (including main tables, and figure legends for main and EV figures) has to be submitted as editable text file (e.g., .docx format). We encourage highlighting of changes (e.g., via text color) for the referees' reference.
- 4) Each main and each Expanded View (EV) figure should be uploaded as individual production-quality files (preferably in .eps, .tif, .jpg formats). For suggestions on figure preparation/layout, please refer to our Figure Preparation Guidelines: <http://bit.ly/EMBOPressFigurePreparationGuideline>
- 5) Point-by-point response letters should include the original referee comments in full together with your detailed responses to them (and to specific editor requests if applicable), and also be uploaded as editable (e.g., .docx) text files.
- 6) Please complete our Author Checklist, and make sure that information entered into the checklist is also reflected in the manuscript; the checklist will be available to readers as part of the Review Process File. A download link is found at the top of our Guide to Authors: embopress.org/page/journal/14602075/authorguide
- 7) All authors listed as (co-)corresponding need to deposit, in their respective author profiles in our submission system, a unique ORCID identifier linked to their name. Please see our Guide to Authors for detailed instructions.
- 8) Please note that supplementary information at EMBO Press has been superseded by the 'Expanded View' for inclusion of additional figures, tables, movies or datasets; with up to five EV Figures being typeset and directly accessible in the HTML version of the article. For details and guidance, please refer to: embopress.org/page/journal/14602075/authorguide#expandedview
- 9) To facilitate reproducibility and cross-laboratory adoption of methodologies, please structure the Materials & Methods section as outlined in our guide to authors, including a completed Reagents and Tools Table that can be downloaded from our author guidelines as well (<https://www.embopress.org/page/journal/14602075/authorguide#structuredmethods>).
- 10) Digital image enhancement is acceptable practice, as long as it accurately represents the original data and conforms to community standards. If a figure has been subjected to significant electronic manipulation, this must be clearly noted in the figure legend and/or the 'Materials and Methods' section. The editors reserve the right to request original versions of figures and the original images that were used to assemble the figure. Finally, we generally encourage uploading of numerical as well as gel/blot image source data; for details see: embopress.org/page/journal/14602075/authorguide#sourcedata

In the interest of ensuring the conceptual advance provided by the work, we recommend submitting a revision within 3 months (5th Feb 2026). Please discuss the revision progress ahead of this time with the editor if you require more time to complete the revisions. Use the link below to submit your revision:

Link Not Available

Referee #2:

All my comments have been fully addressed. It is an exciting study.

Referee #3:

The authors have addressed my concerns and have improved the paper. They have added data showing that LEN treatment stimulates integration by triggering the release of integration competent complexes from CPSF6 clusters. They have also made it clearer that EFV was used to block reverse transcription in certain experiments. The rebuttal letter was convincing.